# Localized Wnt-signaling promotes asymmetric NuMA-dependent oriented divisions and unequal apportioning of mitochondria

Susanna Eli [1,6,7], Greta Rauso [1,7], Paola Ghezzi[1], James L. A. Szczerkowski[2], Michela Bruzzi[1], Francesca Rizzelli[1], Fabiola Iommazzo [1,3], Alessia Loffreda [4], Francesco Castagna [1], Federico Donà [1], Chiara Gaddoni[1], Ambra Dondi[1], Mattia Marenda [1], Simona Rodighiero [1], Pierre Tournier[2], Zeno Lavagnino [5], Dario Parazzoli [5], Nils C. Gauthier [5], Simone Tamburri [1,3], Diego Pasini [1,3], Shukry James Habib [2] ✉ & Marina Mapelli [1] ✉

In multicellular organisms, the execution of developmental and homeostatic programs often relies on asymmetric cell divisions. These divisions require the alignment of the mitotic spindle axis to cortical polarity cues, and the unequal partitioning of cellular components between progeny cells. Asymmetric divisions are orchestrated by signals from the niche frequently presented in a directional manner, such as Wnt signals. Here we employ bioengineered Wnt-niches to demonstrate that in metaphase NuMA/dynein microtubule motors form a complex with activated LRP6 and β-catenin at the cortical sites of Wnt activation to orient cell division perpendicularly. We show that engagement of LRP6 co-receptors by Wnt ligands locally stabilizes actomyosin contractility through the accumulation of myosin1C. Additionally, we describe a proteomic-based approach to identify mitotic protein complexes enriched at the Wnt-contact site, revealing that mitochondria polarize toward localized Wnt3a sources and are asymmetrically apportioned to the Wnt-proximal daughter cell during Wnt-mediated asymmetric cell division of embryonic stem cells. Mechanistically, we show that CENP-F is required for mitochondria polarization towards localized sites of Wnt3a activation, and that deletion of the Wnt-co-receptor LRP6 impairs the asymmetric apportioning of mitochondria. Our findings enhance the understanding of mitotic Wnt-signaling and elucidate fundamental principles underlying Wnt-dependent mitochondrial polarization.

In multicellular organisms, tissue morphogenesis and homeostasis are sustained by stem cell self-renewal that ensures the correct architecture and size of the organs. At a cellular level, self-renewal can occur by asymmetric stem cell divisions that lead to the unequal segregation of cellular components and the differential positioning of daughters compared to niche microenvironments, ultimately resulting in diverse fates[1–3]. Wnt ligands are core components of stem cell niches by virtue of their short-range nature and their capacity to

simultaneously activate lineage-specification and cell-polarizing cascades, both of which are prerequisites for self-renewal[4,5]. The role of Wnt proteins in stemness and differentiation patterning is well-characterized in the mouse small intestine, where stem cells are maintained at the crypt bottom by a Wnt gradient[6,7], and in the specification of hair follicle, which is initiated by an asymmetrical mitotic partitioning of Wnt proteins before a niche is established[8]. The spatial coordination of cell fate specification by Wnt signals remains a subject of ongoing investigation.

Molecularly, secreted lipidated Wnt ligands locally activate Frizzled receptor-mediated signaling to initiate cell polarity and cell fate transduction cascades. Canonical Wnt signaling relies on LRP6/LRP5 coreceptors and leads to β-catenin stabilization and transcriptional activity[9], but also to mitotic Wnt/STOP signaling[10]. Upon Wnt binding, LRP6 coreceptors cluster at the plasma membrane with the scaffold protein Dishevelled-2 (Dvl2), and are phosphorylated by glycogen synthase kinase 3β (GSK3β) and CK1 γ to promote the recruitment of Axin1, adenomatous polyposis coli (APC) and β-catenin[11]. In this way, LRP6/Wnt signals inhibit GSK3β-dependent β-catenin ubiquitination-mediated degradation. Concomitant to this canonical Wnt-transduction pathway, a cell-polarity signaling cascade also exists, acting on cell adhesion and cytoskeletal organization, and relying on the same cytosolic effector Dvl. Intriguingly, in epithelia, the most abundant pool of β-catenin is found at the adherens junctions. Notably, in mitosis, β-catenin, Axin, Dvl2 and APC localize to the spindle poles, and APC associates with spindle microtubules (MTs)[12–15]. In epithelial cells in culture, endogenous Wnt3a signaling is required for integrin-dependent mitotic spindle assembly and alignment to the substratum[16,17]. However, the molecular mechanisms underlying the impact of localized Wnt3a activation in mitosis, and how they contribute to the execution of asymmetric cell divisions (ACDs), are gradually unfolding only recently.

Wnt3a proteins covalently tethered to paramagnetic microbeads (Wnt3a-beads) have been shown to orient the mitotic spindle and induce asymmetric cell divisions of mouse embryonic stem cells (mESCs) and human skeletal stem cells[18,19]. These processes are mediated by components of the Wnt pathway, ionotropic glutamate receptors and atypical PKCζ [20]. These studies imply that a molecular link exists between localized Wnt3a-signaling and microtubule motors orienting the mitotic spindle. In epithelial cells, spindle placement is achieved by recruitment of dynein/dynactin at restricted cortical domains by the activating adaptor nuclear mitotic apparatus protein (NuMA), so that the dynein retrograde motor activity on astral microtubules results in traction forces pulling the spindle toward the cortex[21–24]. In turn, during metaphase, NuMA is targeted to the cortex by direct interactions with the scaffolding protein LGN and the Gαi subunit of heterotrimeric G-proteins[25–27]. Division orientation is also contributed by the contractile actomyosin cortex, which is responsible for mitotic cell rounding and for cortical anchoring of dynein/dynactin via Afadin[28,29]. Whether this evolutionary conserved spindle orientation machinery is involved in Wnt3a-dependent ACDs remains an open question.

Stem cell asymmetric divisions have been associated with unequal partitioning of subcellular components, including signaling molecules, transcription factors, RNAs and organelles, between daughter cells[1,8]. Asymmetric apportioning of mitochondria has been documented[30], and asymmetric partitioning of proteins destined to be degraded has been observed in embryonic stem cells[31]. However, it is not clear whether this unequal organelle partitioning can be instructed by extracellular cues.

Here, we capitalize on the design of artificial niches constructed by covalent immobilization of Wnt3a ligands on paramagnetic beads (Wnt3a-beads) to genetically dissect the pathway instructing division orientation towards the Wnt3a-bead and to establish a general biochemical method for the identification of cellular factors enriched at the Wnt3a-contact site by proteomics. We show that in HeLa cells, Wnt3a-beads polarize the spindle orientation machinery at the contact site, promoting the assembly of NuMA/β-catenin/LRP6 complexes. Mechanistically, Wnt3a-dependent orientation relies on caveolin, Ck1α, and the actin-binding protein MACF1, and induces a local actin-rearrangement sustaining division orientation. In addition, we demonstrate that spatially restricted Wnt3a signals induce the asymmetric distribution of mitochondria toward the Wnt3a bead in a CENP-F-dependent manner. Finally, we demonstrate the capacity of Wnt3a-beads to induce mitochondrial polarization in individual mouse embryonic stem cells (mESCs) and asymmetric apportioning of mitochondria alongside β-catenin levels in post-division cells. Specifically, the Wnt3a proximal cell exhibits a greater proportion of both mitochondria and β-catenin compared to the Wnt3a-distal cell.

## Results

### Localized Wnt3a-sources suffice to orient epithelial cell divisions

To start addressing whether Wnt3a signals can instruct division orientation in epithelial cells, we covalently immobilized Wnt3a on microbeads (Wnt3a-beads) following the previously established protocol[18,19]. The efficacy and reproducibility of the immobilization procedure were verified by TOP-Flash control assays conducted on HEK293T cells stably harboring a 7 × TCF-Luciferase reporter as a readout for canonical β-catenin-dependent transcription (Fig. S1a, b). To evaluate the mitotic response to localized Wnt3a-sources, we performed time-lapse analyses of HeLa cells dividing in contact with a single Wnt3a-bead (Fig. 1a, b and Supplementary Fig. 1c). The orientation of division with respect to the Wnt3a source was evaluated by measuring the angle between the metaphase plate visualized by H2B-GFP and the bead contact site (Fig. 1b, c). These analyses revealed that Wnt3a-beads suffice to orient the mitotic spindle axis, and hence the cell division, perpendicular to the localized Wnt3a source, while bovine serum albumin (BSA) coated-beads cannot. Notably, treatment of Wnt3a-beads with dithiothreitol (DTT), which disrupts the correct folding of the Wnt ligand, abolishes spindle orientation toward the inactive Wnt3a-bead (iWnt3a-bead). Conversely, immobilization of EGF ligands on the beads does not induce division orientation perpendicular to the bead (Fig. 1c and Supplementary Fig. 1d), indicating that the mitotic spindle responds specifically to localized Wnt3a.

In most vertebrate systems, the mitotic spindle first assembles and then, in metaphase, it is centered in the cell and stabilized with an orientation that is maintained through anaphase, and this orientation determines the position of the daughter cells[32]. To monitor the mitotic spindle axis movement of cells dividing in contact with Wnt3a-beads, HeLa cells were synchronized by thymidine block, and Wnt3a-beads were added to the culture 4 h after thymidine wash-out when filming was initiated, which is about 4 h from the metaphase peak of the cell population (Supplementary Fig. 1c). Analyses of metaphase movements revealed that 90% of cells contacting Wnt3a-beads assembled their mitotic spindles already oriented towards the bead (Fig. 1d). This suggests that spatially restricted Wnt3a signals, acting as early as interphase, induce a local rearrangement of the actomyosin cortex. This localized cortical rearrangement anchors the bead at the contact site and persists throughout mitosis, instructing spindle pole positioning at prometaphase, even when the membrane undergoes remodeling during cell rounding.

When plated on fibronectin-coated coverslips, HeLa cells divide with the spindle axis parallel to the substratum in an integrin-dependent manner[33,34]. To explore the crosstalk between forces controlling spindle orientation toward the Wnt3a-bead and adhesive forces guiding alignment to the substratum, we correlated the angle between the spindle axis and the coverslip (Fig. 1e, α-angle) with the angle between the spindle axis and the bead (Fig. 1e, γ-angle). Spindle angle analysis revealed that Wnt3a-beads attract the spindle also along the z-direction, inducing a tilt in the spindle axis compared to the

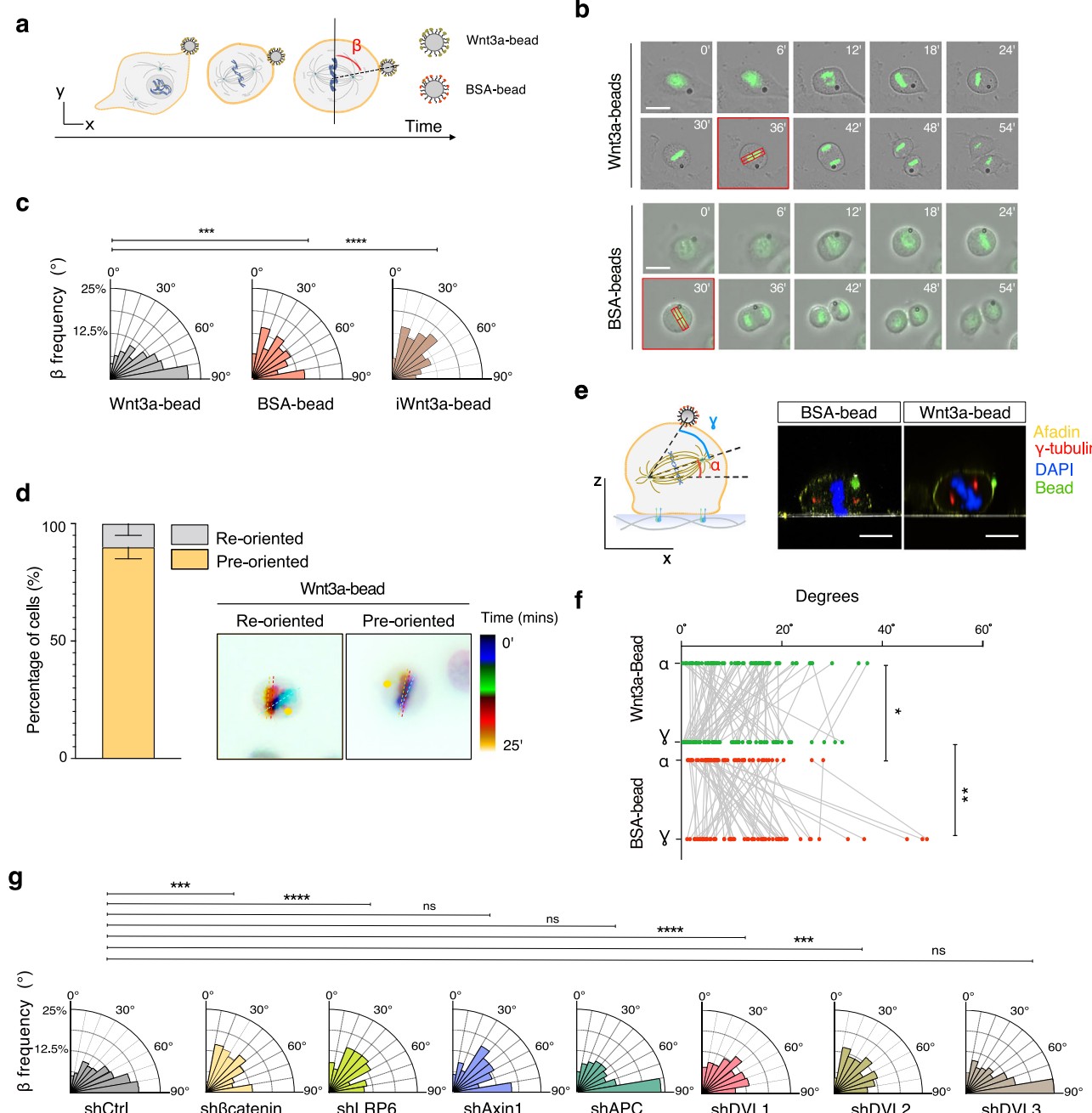

substratum, and imposing an angular spindle axis orientation. This result is not observed with the control BSA-coated beads (Fig. 1e, f) that were also shown to fail to orient the spindle in mESCs[18,19]. Our results suggest that the vertical angular spindle axis distribution is determined by the interplay between cell adhesion forces and localized Wnt3a-induced forces.

To assess the molecular mechanism underlying the Wnt3a-mediated spindle orientation, we utilized HeLa cell lines stably interfered with components of the canonical Wnt-signal transduction cascade, including LRP6, β-catenin, APC, Axin1, Dvl1, Dvl2 and Dvl3 (Supplementary Fig. 1e–m). Time-lapse imaging, conducted using a similar methodology as in wild-type cells, confirmed that LRP6, β-catenin, Dvl1 and Dvl2 are essential for Wnt3a-bead-directed spindle orientation in HeLa cells. Other Wnt pathway components examined did not exhibit this effect (Fig. 1g). Wnt3a signals have been implicated in maintaining the spindle axis parallel to the substratum when cultured epithelial cells divide on adhesive substrates[16]. We next

examined whether the same canonical Wnt components implicated in response to localized Wnt3a sources are also involved in spindle axis alignment to the substratum. Depletion of LRP6, Dvl2, Dvl3 and Axin1 resulted in spindle randomization, with average spindle angles of 13°, 16°, 14° and 12,7°, compared to 7° of control cells (Supplementary Fig. 1n, o). To gain insights into the force balance between Wnt3a-bead orientation and orientation to the substratum, we evaluated the relationship between $\alpha/\gamma$ angles (Fig. 1e) using HeLa cell lines depleted of Dvl3, which misalign the spindle axis compared to the substratum but not to the Wnt3a-bead, or depleted of β-catenin, which retain spindle axis alignment to the substratum but do not orient toward the Wnt3a-bead. This experiment revealed that shDvl3 HeLa cells divide with a tilted spindle axis compared to the substratum even in the presence of the Wnt3a-bead, without losing orientation to the Wnt3a-bead in the z-axis (Supplementary Fig. 1p). In turn, shβ-catenin metaphase HeLa cells retain alignment to the plate and show a wider $\gamma$-angle distribution than shCtrl HeLa cells, though not significant (Supplementary Fig. 1p).

**Fig. 1 | Localized Wnt3a-source suffices to orient epithelial cell divisions.**
**a** Cartoon representation of video-recorded cells dividing in contact with Wnt3a-beads. Cells are seeded with beads before mitosis, and they round up and assemble the mitotic spindle in contact with the beads. Quantification of the spindle axis alignment to the beads was performed by measuring the angle β formed by a line passing through the beads and the middle of the metaphase plate (dotted), and the metaphase plate itself (solid line). **b** Still frames for time-lapse video-recording of HeLa cells stably expressing H2B-GFP dividing in contact with Wnt3a-beads (top series) or BSA-coated beads (bottom series). Beads are visible as dark spheres; the metaphase plate in the frame used for the β-angle measurement is boxed in red. Minutes from the beginning of filming are indicated on the top right corner of each frame. Scale bars, 10 µm. **c** Rose-plots of β-angle frequency of cells imaged in (**b**). The specificity of the orientation was further tested on Wnt3a-beads inactivated by DTT treatment (iWnt3a). Data are shown from 3 independent experiments, with $n = 403$ for Wnt3a-beads, $n = 238$ for BSA-coated beads and $n = 123$ for iWnt3a-beads. The Kruskal–Wallis non-parametric test was applied. ***$p$-value < 0.001, ****$p$-value < 0.0001 (BSA-beads $p$-value = 0.0005). **d** Analysis of the metaphase plate oscillation of HeLa cells expressing H2B-GFP dividing in contact with Wnt3a-beads with a final β-angle between 80°–90°. Images represent a temporal color-coded overlay of the H2B-GFP signal, in which for each time point the metaphase plate is marked by a dashed line (right). The percentage of pre/re-oriented cells was calculated considering a re-orientation event when the metaphase plate rotates more than 10° between two consecutive frames and plotted in bar graphs. Means ± SD are shown. 10 cells per condition were filmed in 3 independent

experiments, taking frames every 5 min (left). **e** Left, scheme of the angles measured in HeLa cells dividing in contact with beads visualized in a z-section: the α-angle (red) is formed by a line passing through the spindle poles and the substratum; the γ-angle (light blue) is formed by a line passing through the spindle poles and the line from the distal pole to the bead. Right, confocal x–z sections of HeLa cells dividing in contact with Wnt3a and BSA-coated beads. Cells were stained with γ-tubulin (red) to visualize the spindle poles, Afadin (in yellow) for the cell cortex and DAPI (blue) for the DNA. Beads are visible by autofluorescence (green), and coverslips are visible in white. Scale bars, 10 µm. **f** Quantification of the relationship between the α and γ angles in metaphase HeLa cells dividing in contact with Wnt3a-beads (top) or BSA-coated beads (bottom). Line plots with the distributions of α and γ angles from 3 independent experiments with $n = 65$ for Wnt3a-beads and $n = 57$ for BSA-beads. The Mann-Whitney non-parametric test was applied. *$p$-value < 0.1, **$p$-value < 0.01 (γ angles $p$-value = 0.0060, α angle $p$-value = 0.0139). **g** Rose-plots of the β angle distribution of HeLa cells depleted of the Wnt effectors β-catenin, LRP6, APC, Axin1, Dvl1, Dvl2 and Dvl3. HeLa cells expressing a scrambled shRNA were used as a negative control (shCtrl), with $n = 768$ from four independent replicates. shβ-catenin $n = 130$, shLRP6 $n = 222$, shAPC $n = 194$, shAxin1 $n = 187$, shDvl2 $n = 165$ and shDvl3 $n = 169$ with three independent replicates; shDvl1 $n = 580$ with four independent replicates. The Kruskal–Wallis non-parametric test was applied. **$p$-value < 0.01; ****$p$-value < 0.0001 (for shβ-catenin $p$-value = 0.0006; for shLRP6 $p$-value < 0.0001; for shAPC $p$-value > 0.9999; for shAxin1 $p$-value = 0.0539; for shDvl1 $p$-value < 0.0001; for shDvl2 $p$-value = 0.0002; for shDvl3 $p$-value > 0.9999).

Wnt signaling is known to inhibit the APC/Axin1-mediated activity of the destruction complex, stabilize β-catenin, and activate the β-catenin-dependent transcription of Wnt target genes, including TCF7 and RNF43[35,36]. Given this, we reasoned that stable downregulation of LRP6, APC or Axin1 by shRNA interference could impact on division orientation mechanism by altering transcription of Wnt targets in interphase. To test this hypothesis, we set out to test whether upregulation of Wnt-mediated transcription by treatment with the GSK3β inhibitor CHIR-99021 in shLRP6 cells could rescue misorientation toward the Wnt3a-beads observed in this cell line (Fig. 1g). We first monitored β-catenin and TCF7 mRNA levels in shLRP6 HeLa cells treated for 4 h with 10 µM CHIR-99021, and found that both were increased (Supplementary Fig. 2a–c). Interestingly, upon CHIR-99021 wash-out and mitotic entry, β-catenin protein levels decrease, whereas TCF7 mRNA remains high. We then checked the effect of CHIR-99021 treatment on division orientation toward Wnt3a-beads, and found that it neither randomized the orientation of shCtrl cells nor altered the misorientation observed in shLRP6 cells (Supplementary Fig. 2d). Next, we investigated whether shAPC and shAxin1 HeLa cells stabilize β-catenin and upregulate Wnt target genes transcription, as expected upon impairment of the destruction complex, and found that only Axin1 depletion induces a significant increase in β-catenin and TCF7/RNF43 mRNA levels (Supplementary Fig. 2e, f). Thus, we decided to use the shAxin1 cells to test whether inhibition of β-catenin-mediated transcription could impact division orientation toward the Wnt3a-bead by treating them with 10 µM LF3, which disrupts the β-catenin/TCF4 interaction[37] (Supplementary Fig. 2g). This experiment showed that inhibition of Wnt transcription targets does not alter β-angle distribution of shAxin1 HeLa cells toward Wnt3a-beads (Supplementary Fig. 2h).

Collectively, these findings indicate that in epithelial cells in culture, localized Wnt3a-signals suffice to instruct division orientation toward the source by breaking the symmetry of the cell cortex at the contact site. This process impinges on the Wnt components LRP6, β-catenin and Dvl2, is coordinated with adhesive forces, and is not affected by Wnt gene transcription.

**NuMA/LGN/Gαi spindle orientation complexes are recruited at the Wnt3a-bead contact site**
In vertebrate cells, spindle placement is attained by conserved NuMA/LGN/Gαi complexes that target to the cortex dynein/dynactin motors,

whose minus-end-directed movement on astral MTs results in pulling forces rotating the spindle[22,24]. To explore whether the same mechanism is responsible for spindle orientation towards localized Wnt3a-sources, we examined the division orientation towards Wnt3a-beads of HeLa cells stably depleted of NuMA[38], LGN[28], Gαi1 or Gαi3 (Supplementary Fig. 3a, b), and found that loss of any of these proteins disrupts orientation perpendicular to the bead (Fig. 2a). In turn, stable depletion of the actin-binding protein Afadin, which has been shown to contribute to spindle alignment to the substratum[28], does not perturb the mitotic spindle response to Wnt3a-signals (Fig. 2a).

We next built on this genetic evidence to investigate the mechanistic basis of the mitotic spindle response to Wnt3a-beads. We imaged the distribution of Gαi1, LGN, NuMA and the dynactin subunit p150-Glue in HeLa cells dividing in contact with Wnt3a-beads, and found that in metaphase the dynactin/NuMA/LGN/Gαi complex distributes in crescents above the spindle poles, and that one of these crescents is centered at the bead contact site where these proteins accumulate (Fig. 2b, c and Supplementary Fig. 3c–f). In line with the evidence that BSA-coated beads cannot instruct spindle orientation, no enrichment of NuMA, LGN, Gαi and p150-Glue is observed at BSA-bead contact sites (Fig. 2c).

Mitotic spindle placement is orchestrated by traction forces exerted by dynein/dynactin motors on astral MTs emanating from the spindle poles. Consistent with this notion, in HeLa cells dividing in contact with Wnt3a-beads, astral MTs extend from the bead-proximal pole to the cortex in contact with the bead (Fig. 2e and f). Depolymerization of astral MTs by low doses of nocodazole disrupts HeLa cells' division orientation toward Wnt3a-beads (Fig. 2d), indicating that they are implicated in Wnt3a-induced spindle placement. Together, these findings demonstrate that localized Wnt3a activation orients cell divisions by hijacking the spindle orientation machinery assembled on dynactin/NuMA/LGN/Gαi complexes at the cortically localized source of Wnt3a signals and relies on dynein-based astral microtubule pulling forces.

**Mitotic proteome recruited at Wnt3a sources contact sites**
Our data show that localized Wnt3a sources induce the accumulation of NuMA/LGN/Gαi at the bead contact site. This prompted us to search for additional components recruited to the Frizzled/LRP6 receptors locally activated by Wnt3a engagement, possibly casting more light on the molecular pathways triggered by mitotic Wnt3a signals. To this

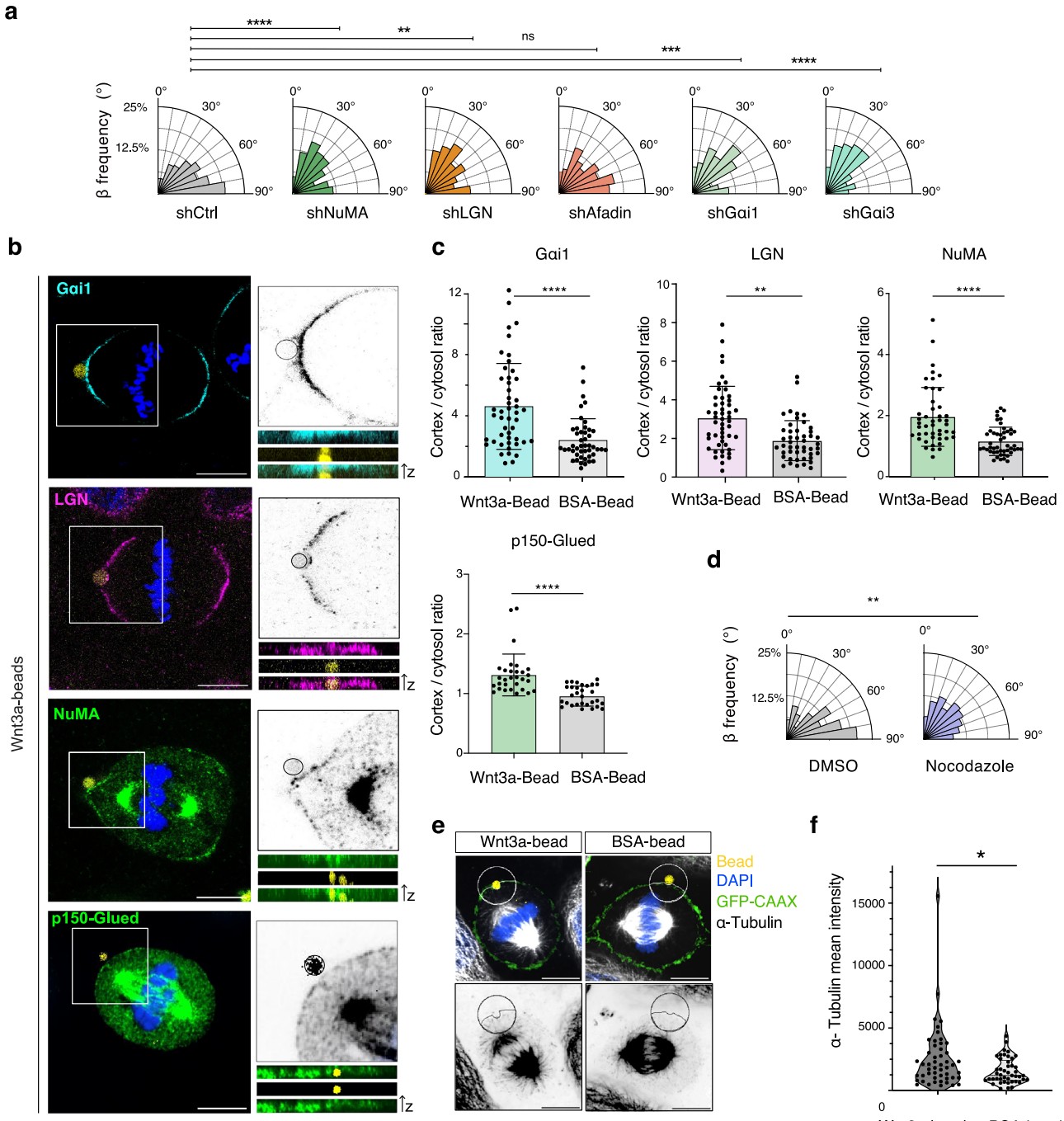

**Fig. 2 | The NuMA/LGN/Gαi spindle orientation pathway is recruited at the Wnt3a-contact site. a** Rose-plots of the *β*-angle distribution of HeLa cells depleted of the spindle orientation proteins NuMA, LGN, Afadin, Gαi1 and Gαi3. *β*-angle distribution is shown for three independent experiments, with shCtrl *n* = 337, shNuMA *n* = 346, shLGN *n* = 308, shAfadin *n* = 308, shGαi1 *n* = 143 and shGαi3 *n* = 92. The Kruskal–Wallis non-parametric test was applied. **p-value < 0.01, ***p-value < 0.001, ****p-value < 0.0001 (shLGN *p*-value = 0.0017; shAfadin > 0.99, shGαi1 *p*-value = 0.0004). **b** Confocal images of mitotic HeLa cells dividing in contact with Wnt3a-beads (yellow) stained for p150-Glued (green) NuMA (green), LGN (magenta) and Gαi1 (cyan), with a black-and-white magnification on the right. Scale bar, 10 μm. **c** Quantification of the cortical/cytosol signal intensity ratio of p150-Glued, NuMA, LGN and Gαi1 of HeLa cells in panel **b** for Wnt3a-coated and BSA-coated beads. Unpaired *T*-test was performed for three independent experiments with p150-Glue Wnt3a-bead *n* = 30; BSA-bead *n* = 30; NuMA Wnt3a-bead *n* = 46; BSA-bead *n* = 48; LGN Wnt3a-bead *n* = 51; BSA-bead *n* = 48; Gαi Wnt3a-bead

*n* = 53; BSA-bead *n* = 46. **p-value < 0.01, ****p-value < 0.0001 (*p*-value LGN = 0.0014). Means ± SD are shown. **d** Rose plots of β-angle distributions of HeLa cells treated with 20 nM nocodazole or with 0.05% DMSO as a negative control from three independent experiments, *n* = 142 in DMSO and *n* = 152 in nocodazole treatments. The Mann-Whitney non-parametric test was applied. **p-value < 0.001 (*p*-value = 0.0017). **e** Confocal images of metaphase HeLa cells in contact with a Wnt3a-bead (left, yellow) or BSA-bead (right, yellow) stained for α-tubulin (white), DNA (DAPI, blue), and the plasma membrane (GFP-CAAX, green). Lower panels show inverted grayscale images of the α-tubulin signal. For each condition, images were taken from three independent experiments, with cell numbers indicated in (**f**). Scale bar, 10 μm. **f** Violin plot of the distribution of the mean intensity of the α-tubulin signal in proximity of the beads for each condition. Data are shown from three biological replicates, with Wnt3a-beads *n* = 50 and BSA-beads *n* = 45. Statistical analysis was performed using an unpaired *t*-test. *p-value < 0.05 (*p*-value = 0.0175).

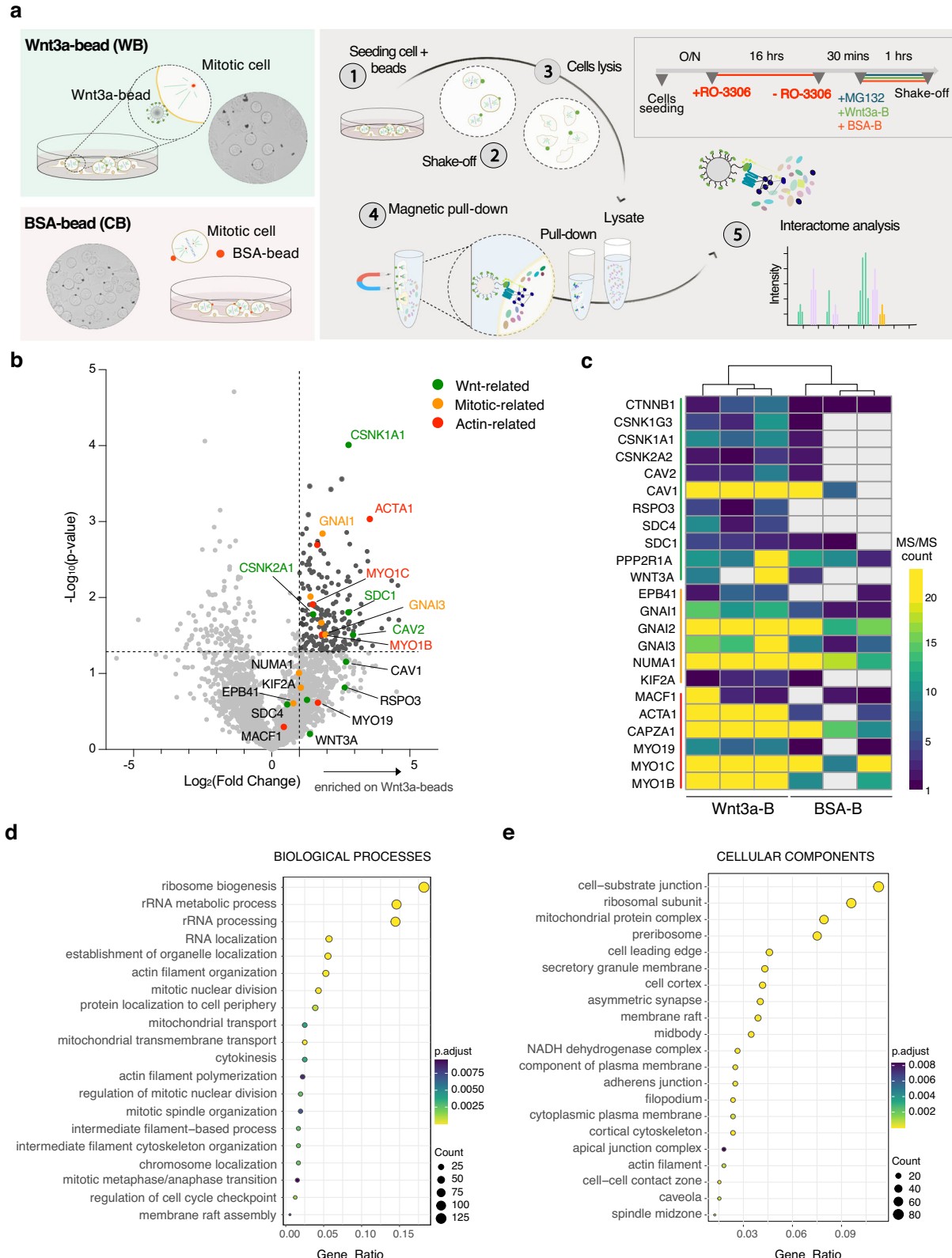

aim, we developed a proteomics-based approach to isolate membrane-associated and cytosolic factors recruited at the Wnt3a-bead contact site, and identified them by mass-spectrometry. The protocol takes advantage of the paramagnetic properties of the microbeads used for Wnt3a immobilization, which allow to separate them from the lysate without clearing. Briefly, HeLa cells were synchronized in metaphase and seeded with Wnt3a or BSA-coated beads (Fig. 3a, top right).

Rounded metaphase cells were harvested by shake-off, lysed in the presence of 0.2% NP-40, and protein complexes adsorbed on microbeads were magnetically isolated for mass-spectrometry identification (Fig. 3a). In total, we identified 439 proteins significantly enriched at the Wnt3a-beads compared to BSA-coated beads, underscoring the efficacy of the isolation method. A number of Wnt effectors were reproducibly enriched at the Wnt3a-beads compared to BSA

**Fig. 3 | Mitotic proteome recruited at Wnt3a-source contact sites. a** Principles of the magnetic bead pull-down experiment used for the identification of the proteome recruited at the Wnt3a contact site in metaphase HeLa cells. Left, cartoon representation and brightfield images of rounded HeLa cells synchronized in metaphase and seeded with beads (dark spheres). Right: Workflow of the proteomic experiment. HeLa cells were synchronized in G2/M by RO-3306 treatment, released in fresh medium, and subsequently supplemented with MG132 to arrest them in metaphase. Wnt3a/BSA-coated magnetic beads were added concomitantly to MG132 (1). Cells were harvested after 1 h by shake-off (2), lysed in 0.2% NP-40 (3), and cellular components enriched at the beads, including extracellular factors, plasma-membrane-associated proteins and cytoplasmic proteins, were isolated from the lysate by magnetic pull-down (4). The composition of the complexes

enriched at the beads was analyzed by LC-MS/MS, comparing results obtained for Wnt3a-coated and BSA-coated beads. **b** Volcano plot of pull-down mass spectrometry data (Wnt3a-beads versus BSA-coated beads). *T*-test difference of LFQ values between the two conditions is plotted on the *X*-axis as Log$_2$(Fold change), while significance is plotted on the *Y*-axis as −Log10(*t*-test *p*-value). Proteins significantly pulled-down by Wnt3a-beads are colored in black, and further labeled in green (Wnt-related), yellow (mitotic-related) and red (actin-related) according to their functional category. **c** Heatmap of spectral counts of selected proteins. Spectral count (MS/MS) is plotted for each of the three replicates per condition. **d** and **e** Functional analysis of significantly interacting proteins. The 20 most relevant GOs are selected among the enriched terms for both biological processes (**d**) and cellular components (**e**).

control beads (Fig. 3b, c and Supplementary Table 1). These included the kinases CK1α and CK2α, β-catenin, caveolins, R-spondin-3 and Wnt3a itself. The PP2A phosphatase regulator PPP2R1A and the transmembrane proteoglycan syndecan-1, already known to be involved in Wnt-signal transduction and planar cell polarity pathways[9,39,40], were also found among the most significant hits. Notably, several spindle orientation proteins displayed high abundance at Wnt3a-beads, such as Gαi1 and Gαi3, NuMA, and the MT depolymerase Kif2A, recently shown to respond to global mitotic Wnt3-signaling in RPE cells treated with the Wnt inhibitor Dickkopf-1 (DKK1)[41,42]. More interestingly, the actin-related protein myosin1B, myosin1C and MACF1 were specifically retained on Wnt3a-beads, in agreement with previous observation that Wnt3a locally remodels the actomyosin cortex and triggers cytoskeletal rearrangements essential for division orientation (Fig. 1d)[20,43].

Ontology analysis of Wnt3a-dependent local proteome of the enriched components, revealed prominent classes that are associated with RNA processes, including RNA localization and ribosomes (Fig. 3d, e and Supplementary Fig. 4a). In addition, beside protein families associated to mitotic progression, actin organization, and membrane dynamics, the Wnt3a proteome contains several mitochondrial components and proteins related to mitochondrial metabolism (Fig. 3d, e and Supplementary Fig. 4b). These findings may suggest a correlation between localized Wnt3a signaling and mitochondrial activity or distribution. Although we cannot exclude that these RNA-related and mitochondrial factors have mitotic moonlighting functions not yet annotated, their presence suggests that Wnt3a-beads can instruct the asymmetric distribution of RNA-related proteins and mitochondria. This distribution might promote their unequal apportioning between daughter cells at cytokinesis. In summary, localized Wnt3a signals physically recruit Wnt-transduction components and spindle orientation complexes, generating pulling forces on astral MTs to orient the mitotic spindle. Moreover, at the Wnt3a contact site, RNA-binding proteins and mitochondrial components are enriched.

## Upon localized Wnt signaling activation, the dynein adaptor NuMA physically interacts with LRP6

To validate the presence of MT motors at the Wnt3a-bead contact site, we performed magnetic bead pull-down assays in which we isolated components adsorbed on beads, similarly to what we had done for proteomic analysis, followed by immunoblotting. These experiments confirmed that localized Wnt3a ligands engage specifically LRP6 co-receptors, whose phosphorylated intracellular domain recruits Dvl2 and β-catenin (Fig. 4a and Supplementary Fig. 5a). NuMA is also found enriched at the Wnt3a-beads, together with the spindle orientation proteins LGN and Gαi. In epithelial cells, NuMA promotes spindle orientation by recruiting dynein/dynactin at a locally restricted cortical area. Immunoblot of the dynactin subunit p150$^{Glue}$ revealed that productive NuMA/dynein/dynactin force generators are targeted to the Wnt3a-bead sites (Fig. 4a).

The observation that components of the spindle orientation machinery and Wnt-signaling colocalize at the Wnt3a-bead contact site opens the possibility that they could be part of the same macromolecular complex. To address this possibility, we performed immunoprecipitation (IP) experiments in mitotic HeLa cells stably expressing low levels of LRP6 treated with Wnt3a-conditioned medium (Wnt-ON) or with DKK1-conditioned medium (Wnt-OFF), which inhibits Wnt3a-signaling[42]. Under these conditions, LRP6 enters a complex with β-catenin and NuMA only in Wnt-ON conditions (Fig. 4b). Upon binding to Wnt3 ligands, the intracellular domain of LRP6 is phosphorylated on five PPPSPxS motifs by CK1 and GSK3β, and this is required for activation of the Wnt signal transduction cascade[44–47]. To test whether LRP6 phosphorylation is required for NuMA interaction, we generated an LRP6-M5 mutant in which we replaced the Ser/Thr residues of the five PPPSPxS with alanine, and assessed its ability to bind NuMA. An IP experiment conducted in metaphase HeLa cells in Wnt-ON conditions showed that LRP6-M5 is unable to enter a complex with NuMA or with β-catenin, indicating that activation of the coreceptor at the cortex is required for the interaction with NuMA (Fig. 4c). We reasoned that the LRP6/NuMA interaction could favor the cortical recruitment of microtubule motors organized on NuMA at the site of Wnt signal activation. To test this hypothesis, we evaluated the cortical levels of NuMA in proximity to the Wnt3a-beads in shLRP6 HeLa cells upon transfection of LRP6 rescue constructs. Under these conditions, in shCtrl HeLa cells, NuMA is enriched at the Wnt3a-bead, while in cells lacking LRP6, which do not orient the mitotic spindle toward the Wnt3a-bead, this localization is lost (Fig. 4d and e). Transient transfection of wild-type LRP6 rescues cortical NuMA levels at the Wnt3a-beads, whereas transfection of LRP6-M5 does not (Fig. 4d and e and Supplementary Fig. 5b). Collectively, these findings revealed that during mitosis localized Wnt3a signaling triggers the formation of cortical phospho-LRP6/NuMA/β-catenin complexes, which are essential for cortical recruitment of dynein/dynactin microtubule motors promoting division orientation orthogonal to the site of Wnt activation.

## Localized Wnt3a source orients division by local remodeling of membrane and actomyosin cortex

Sustained spindle movements and traction forces are required for spindle elongation and sister chromatid separation. At mitotic entry, the actin cytoskeleton undergoes a major reorganization resulting in the formation of a rigid actomyosin cortex underlying the plasma membrane, which is responsible for mitotic cell rounding[48]. In cultured cells, at mitotic rounding, caveolin-1 is targeted to cortical regions proximal to retraction fibers and organizes caveola-like structures, recruiting NuMA/LGN/Gαi complexes to orient the division axis[49]. We found caveolin-1 and caveolin-2 enriched in our Wnt3a-bead proteomic analysis (Fig. 3b, c and Supplementary Table 1). To assess whether caveolins are involved in transducing localized Wnt3a signaling to the spindle orientation machinery, we first confirmed that caveolin-1 is recruited at Wnt3a-beads by magnetic pull-down assay and imaging

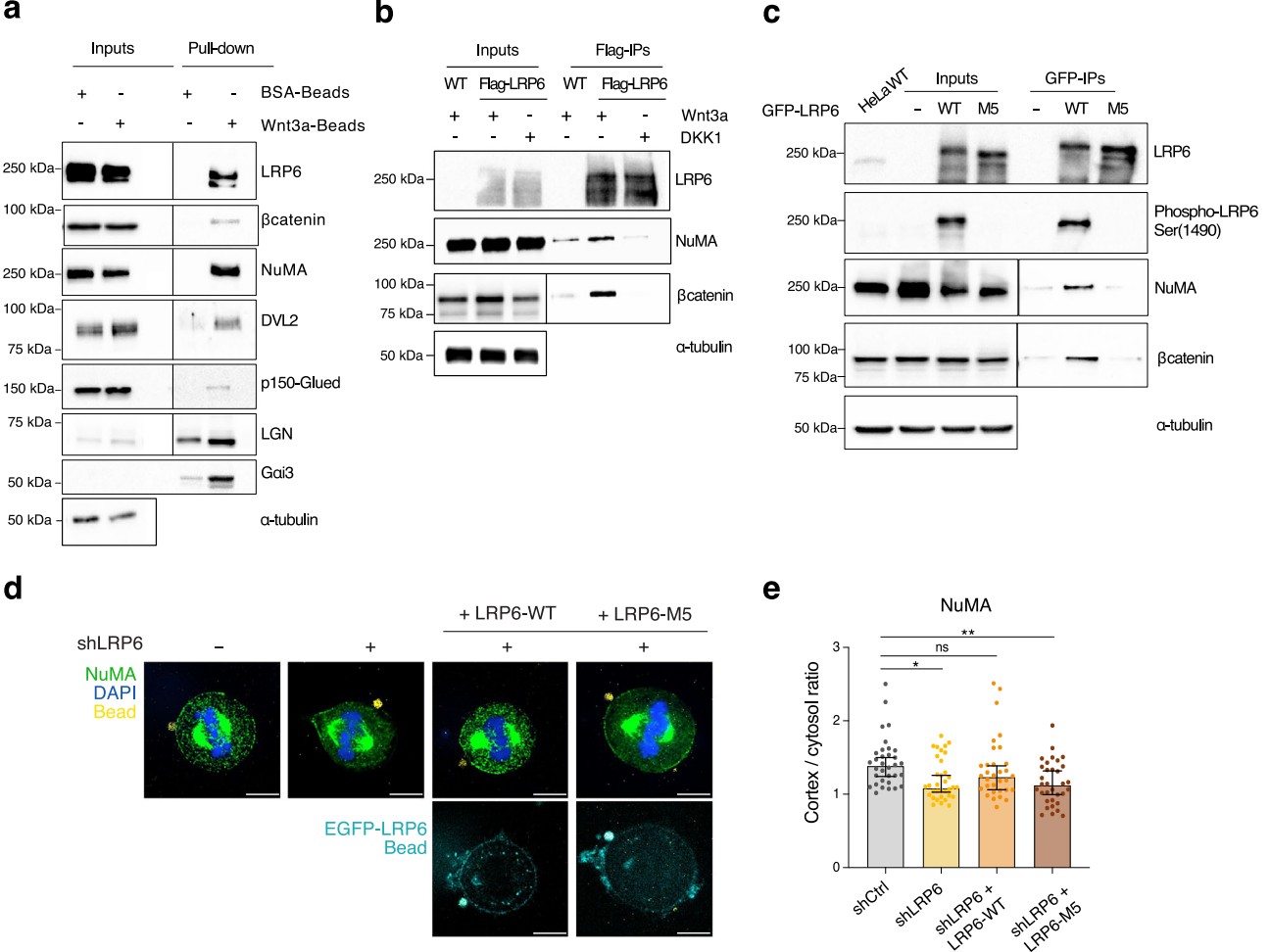

**Fig. 4 | NuMA interacts with LRP6 and β-catenin at localized Wnt3 sources.**
**a** Pull-down assay with Wnt3a-coated or BSA-coated magnetic beads was conducted with HeLa cells synchronized in metaphase. After cell lysis, species retained on beads were separated on SDS-PAGE and immunoblotted with the indicated antibodies. 20 μg of mitotic lysates were loaded as input, and α-tubulin was used as a loading control. The experiment was repeated at least three times. **b** Upon Wnt activation, cortical LRP6 associates with NuMA. HeLa cells stably expressing Flag-LRP6 were treated with Wnt3a-CM or DKK1-CM and synchronized in metaphase. Mitotic lysates were immunoprecipitated with anti-Flag antibodies, and species retained on beads were blotted with the indicated antibodies. Analogous IP experiments were conducted on wild-type (WT) HeLa cells as a specificity control. α-tubulin was used as a loading control. The experiment was repeated three times. **c** shLRP6 HeLa cells transfected with sh-resistant EGFP-LRP6 wild-type (WT) or

mutated in the five PPPSPxS phosphorylation motifs (M5 mutant) were stimulated with Wnt3a-CM, and synchronized in metaphase. Mitotic lysates were subjected to immunoprecipitation with anti-GFP beads, and bound proteins were analyzed by immunoblotting with the indicated antibodies. Wild-type HeLa cells were used as a specificity control. α-tubulin was included as a loading control. The experiment was repeated at least three times. **d** Representative IF of rescue experiments of cortical NuMA (green) at the Wnt3a-bead contact site in HeLa cells depleted of endogenous LRP6 and transfected with sh-resistant EGFP-tagged LRP6 rescue constructs. Bottom, localization of transfected GFP-LRP6. Scale bar, 10 μm. **e** Quantification of cortical NuMA signal at the Wnt3a-beads in three independent rescue experiments with shCtrl $n = 33$, shLRP6 $n = 34$, shLRP6 transfected with LRP6-WT $n = 32$; shLRP6 transfected with LRP6-M5 $n = 33$. Ordinary One-Way ANOVA between conditions was performed, *$p$-value = 0.0144; **$p$-value = 0.0023. Means ± SD are shown.

(Fig. 5a and Supplementary Fig. 5c). Then we examined the functional relevance of this localization for division orientation by generating a HeLa cell line stably depleted of caveolin-1 (Supplementary Fig. 5d). Caveolin-1 and caveolin-2 act as heterodimers, and ablation of caveolin-1 destabilize their assembly resulting in loss of caveolin-2 as well (Supplementary Fig. 5d)[50]. Time-lapse microscopy showed that division orientation towards Wnt3a-beads of sh-cav1 HeLa cells is randomized compared to control cells (Fig. 5b), indicating that caveolins play a key role in Wnt3a-dependent orientation.

Spindle orientation relies on the coordination between mitotic actin and MT cytoskeletons, which needs to be specifically tuned in response to localized Wnt3a stimuli. MACF1, enriched at the Wnt3a-bead contact site (Fig. 3b and c), was previously shown to polarize MTs along F-actin focal adhesion cables during directional skin stem cell migration driven by Wnt3a signal activation[51]. We investigated whether in mitosis, MACF1 could stabilize astral MT to the cortex at the Wnt3a-

bead contact site. We first confirmed that MACF1 enriches at the Wnt3a-bead by magnetic pull-down (Fig. 5a). We next showed that MACF1 downregulation by shRNA interference in HeLa cells randomizes the division orientation (Fig. 5b and Supplementary Fig. 5e). Importantly, both caveolins and MACF1 specifically respond to Wnt3a stimuli, implying that a signal transduction cascade instructs their spindle orientation functions.

The top hit in our Wnt3-bead proteomics is CK1α (Fig. 3b, c and Supplementary Table 1), a major Wnt kinase phosphorylating β-catenin on Ser45, whose enrichment at Wnt3-bead contact site was confirmed by magnetic bead pull-downs (Fig. 5a). To explore the functional implications of CK1α recruitment at the beads, we generated HeLa cells stably depleted for CK1α by shRNA interference (Fig. S5f), or treat HeLa cells with the recently characterized CK1 inhibitor MU1742, which also inhibits CK1α[52] (Supplementary Fig. 5g). We found that in both conditions, cells lost the perpendicular alignment to the Wnt3a-bead

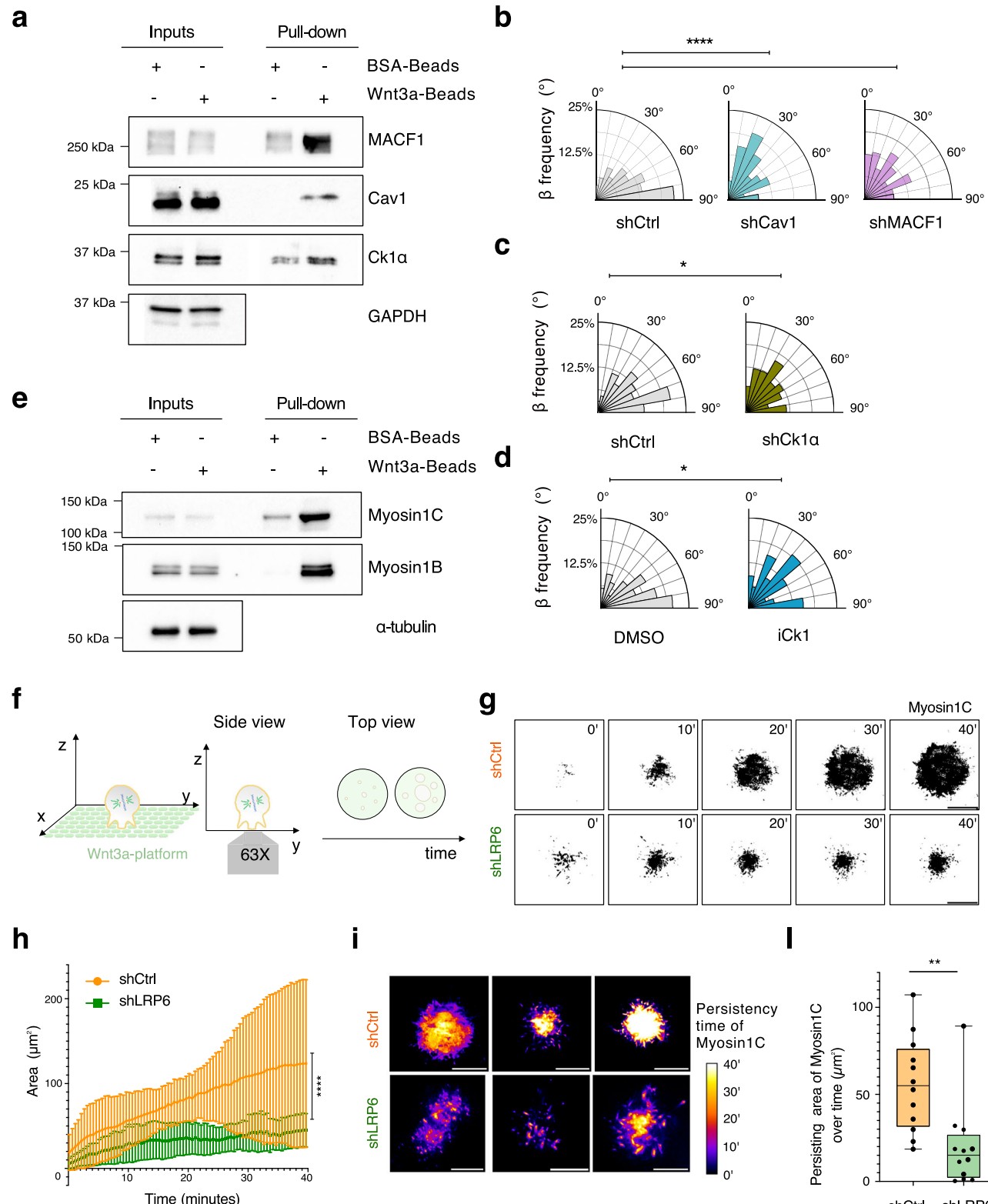

(Fig. 5c and d). Based on these findings, we concluded that Wnt3a locally triggers CK1α kinase activity required for spindle orientation. The Wnt3a-bead proteome was enriched in actin-binding proteins, including the two class 1 myosins, myosin1B and myosin1C. These single-head monomeric actin-based motors can bind to the actin cytoskeleton through their motor domains and to the plasma membrane through their tails, and have been associated with adhesion and control of membrane tension. We then reasoned that myosin1B and

myosin1C could participate in local rearrangement of actomyosin contractility at the site of localized Wnt3a activation in order to strengthen locally the cortex to counterbalance pulling-forces orienting the spindle. To test this hypothesis, we first confirmed that myosin1C and 1B are recruited at Wnt3a-beads by magnetic pull-down assay (Fig. 5e), and then set out to image myosin1C in mitotic cells landing on Wnt3a-coated platform by Total Internal Reflection Fluorescence (TIRF) microscopy (Fig. 5f and Supplementary Fig. 5i, j). HeLa cells were

**Fig. 5 | Mitotic Wnt-signals trigger local actomyosin rearrangements promoting adhesion. a** Pull-down assay with Wnt3a-coated or BSA-coated magnetic beads was conducted with HeLa cells synchronized in metaphase. After cell lysis, species retained on beads were separated on SDS-PAGE and immunoblotted with the indicated antibodies. 20 µg of mitotic lysates were loaded as input, and GAPDH was used as a loading control. The experiment was repeated three times. **b** Rose plots of the β angle distribution of HeLa cells depleted of the caveolin-1 and MACF1 protein. β-angle distribution is shown for three independent experiments, with shCtrl $n = 271$, shCaveolin-1 $n = 151$, shMACF1 $n = 121$. The Kruskal–Wallis non-parametric test was applied. ****$p$-value < 0.0001. **c** Rose plots of β-angle distribution shown for three independent experiments, with shCtrl $n = 130$, shCk1α $n = 139$. The Kruskal–Wallis non-parametric test was applied. *$p$-value < 0.1 ($p$-value = 0.0118). **d** Rose plots of $β$-angle distribution shown for three independent experiments of HeLa cells treated with 10 µM of the Ck1 inhibitor MU1742, with $n = 135$, or 0.5 % DMSO as control $n = 162$. The Kruskal–Wallis non-parametric test was applied. *$p$-value < 0.1 ($p$-value = 0.0238). **e** Pull-down assay with Wnt3a-coated or BSA-coated magnetic beads. After cell lysis, myosin1C and myosin1B absorption on beads was analyzed by SDS–PAGE. α-tubulin was used as a loading control. The

experiment was repeated three times. **f** Cartoon representation of the spreading assay of HeLa cells transfected with myosin1C-YFP on Wnt3a-coated platforms. The area of the myosin1C TIRF signal was analyzed over time. **g** Inverted-color TIRF images of the YFP-myosin1C signal over time on the Wnt3a surface. shCtrl HeLa cells, on the top, were compared with shLRP6 HeLa cells on the bottom. Scale bar, 10 µm. **h** Area of the YFP-myosin1C signal in time for cells spreading on the Wnt3a-platform for shCtrl control cells and shLRP6 cells. Bars identify the average area and its standard deviation for every time point calculated for three independent experiments with $n = 11$. Two-tailed $t$-test was performed for all time points with ****$p$-value < 0.0001. **i** Time-stack projection of 6 representative cells showing the stability of the myosin1C engrafted contact points on the surface. Every pixel value represents the time for which the myosin1C was positively detected (see the "Methods" section for details). Scale bar, 10 µm. **l** Persistence analysis of myosin1C. This quantity was calculated as the area of the myosin1C signal persisting for at least 22.5 min (around half of the time lapse duration). The analysis was performed for three independent experiments with $n = 12$. Two-tailed $t$-test was performed with **$p$-value < 0.01 ($p$-value = 0.0024).

transfected with YFP-myosin1C (Supplementary Fig. 5h), synchronized in mitosis and spreading assays were conducted to monitor the adhesion of mitotic cells upon Wnt3a ligand engagement with the Frizzled/LRP6 receptors. In the 40 min of TIRF imaging, the overall myosin1C area contacting the Wnt3a-platform steadily increases (Fig. 5g and h), with stable attachment of initial contact points for the entire duration of the time-lapse (Fig. 5i–l and Supplementary Movies 1 and 2). To understand whether spreading of cells on the Wnt3a-coated platforms was dependent on Wnt3a-engagement, we evaluated the dynamics of adhesion on the platform of HeLa cells stably depleted of LRP6 by lentiviral shRNA transduction (Supplementary Fig. 1f). This comparison revealed that loss of LRP6 reduces the area of cell adhesion to the Wnt3a-coated platform and prevents the formation of durable receptor engagements (Fig. 5g–l). We conclude that in mitotic cells, myosin1C is recruited to the sites of Wnt3a-dependent LRP6 activation to locally strengthen adhesion by modeling actomyosin contractility, and to sustain spindle orientation and possibly organelle transport toward the site of Wnt activation.

## Localized Wnt3a-activation induces asymmetric distribution of mitochondria in mitosis

Oriented division of stem cells resulting in unequal placement of daughters with respect to the niche often results in asymmetric cell fate contributed by the diverse cellular microenvironment, but also by differential partitioning of cellular components. GO-enriched analysis of the mitotic Wnt3a-bead proteome revealed that the prominent classes are linked to RNA/ribosome processes and mitochondrial transport (Fig. 3d, e and Supplementary Fig. 4a, b). Validation of selected hits belonging to the mitochondrial-GO terms by bead pull-down assays confirmed the bead-proximity of myosin-19, implicated in mitochondrial transport (Supplementary Fig. 6c and d)[53,54]. To gain better insights into the biological relevance of these interactions, we imaged metaphase HeLa cells dividing in contact with Wnt3a-beads by electron microscopy, at a resolution that allows untangling of the complexity of the mitochondrial network. In line with the notion that cell division is oriented perpendicularly towards the Wnt3a-bead, in cells in contact with Wnt3a-beads, the metaphase plate is visible equatorially below the bead (Fig. 6a). Electron micrographs show that the mitochondrial density significantly increased in the proximity of the Wnt3a-bead compared to the opposite site of the cell and to other sectors of comparable area. This mitochondrial enrichment was not observed close to BSA-beads (Fig. 6a, b and Supplementary Fig. 6a, b). Overall, our results demonstrate that in metaphase, mitochondria polarize toward the Wnt3a-bead contact site.

We then set out to dissect the molecular requirements for the mitochondrial polarization towards the Wnt3a-beads observed in

metaphase HeLa cells. To this end, we first developed a quantification protocol to evaluate the mitochondrial density in concentric shells centered at the bead centroid starting from a 3D reconstruction of the mitochondrial network (Fig. 6c and d), and showed that mitochondrial density is higher in proximity to Wnt3a-beads compared to BSA-beads (Fig. 6e), with a median proximal-to-distal mitochondrial density ratio of 1.71 for Wnt3a-beads and 1.35 for BSA-beads (Fig. 6f). We then evaluated how the mitotic mitochondrial distribution changes in cells depleted of key proteins implicated in mitochondrial transport. Recent evidence revealed that the coiled-coil protein CENP-F assists mitochondrial movements to the cell periphery along microtubule tracks by bridging between microtubule plus-ends and the mitochondrial transmembrane GTPases Miro1 and Miro2[54,55]. Interestingly, CENP-F and Miro2 were found enriched in the Wnt3a-beads proteomic (Supplementary Fig. 6c). To assess whether CENP-F and Miro proteins are implicated in mitochondrial polarization toward the Wnt3a-beads, we generated HeLa cells depleted of CENP-F or doubly depleted of Miro1 and Miro2 (Supplementary Fig. 6e and f). Based on measurements of the proximal-to-distal mitochondrial density ratio, metaphase shCENP-F HeLa cells do not polarize mitochondria to the Wnt3a-beads as effectively as the control cells (Fig. 6g–i). In turn, stable depletion of Miro1 and Miro2 reduces cell viability, induces division misorientation towards the Wnt3a-beads (Supplementary Fig. 6g), and randomizes the proximal-to-distal mitochondrial density ratio to the Wnt3a-beads in the subset of oriented cells (Supplementary Fig. 6h–j). Collectively, these findings indicate that in mitosis, localized Wnt signals polarize the mitochondrial distribution towards the site of Wnt activation by triggering directional CENP-F-mediated transport on microtubules.

## Wnt3a-sources instruct the asymmetric apportioning of mitochondria in mESCs

We then investigated whether mitochondrial enrichment could be observed in other cell types and how mitochondria are distributed post-division. To investigate this, we utilized a mouse embryonic stem cell (mESC) model of Wnt-mediated asymmetric cell division. In this system, Wnt3a-coated beads polarize components of the Wnt/β-catenin pathway, including β-catenin, to the contact site. This polarization orients the mitotic spindle towards the bead, inducing asymmetric cell division. Consequently, two distinct daughter cells arise: a Wnt-proximal cell and a Wnt-distal cell. The Wnt-proximal daughter cell displays elevated expression of pluripotency markers and Wnt/β-catenin pathway components. In contrast, the Wnt-distal daughter cell downregulates these components while upregulating differentiation markers, suggesting a fate shift towards an epiblast stem cell lineage[18]. To analyze mitochondrial distribution in this system, we exposed single mESCs to Wnt3a-coated beads. Mitochondrial localization was

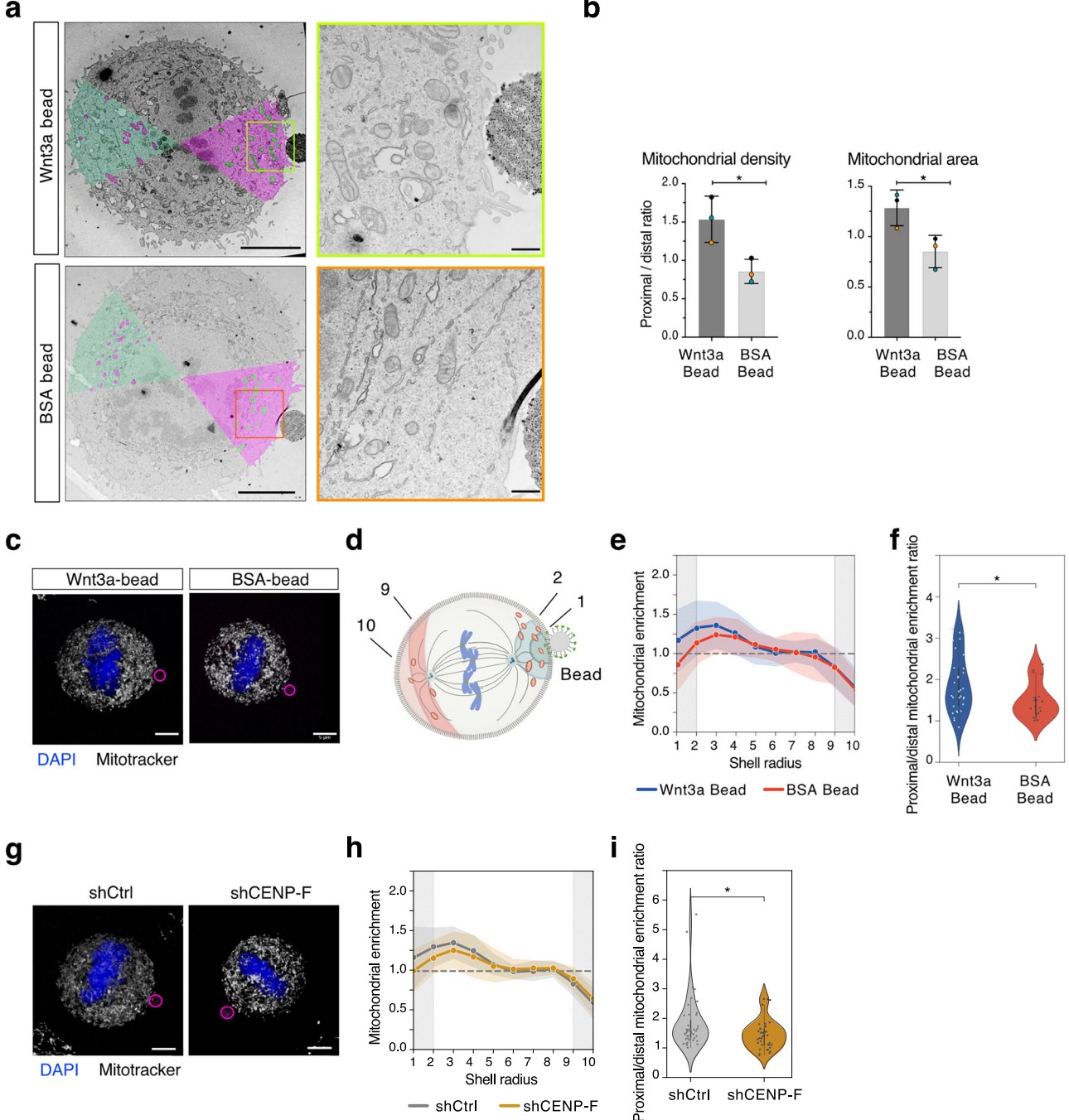

visualized via immunofluorescence using the outer mitochondrial membrane receptor, Tom20 and 3D imaging, followed by quantitative analysis. We observed a significant increase in mitochondrial density at the contact site between the mESC and the Wnt3a-bead compared to other regions within the cell and relative to those exposed to inactive (i) Wnt3a-beads (Fig. 7a, b and Supplementary Fig. 7). Furthermore, post-division analysis revealed a preferential enrichment of both mitochondria and β-catenin in Wnt3a-proximal daughter cells (Fig. 7c−e).

We previously demonstrated that mESCs lacking the Wnt co-receptor LRP6 (LRP6KO) exhibit a 6% basal level of Wnt/β-catenin signaling, likely due to the presence of LRP5[20]. Unlike wild-type mESCs, LRP6KO cells fail to orient their mitotic spindles towards Wnt3a-beads, and the asymmetric distribution of pluripotency markers (Nanog, Rex1, Sox2) between daughter cells is impaired[20], indicating a disruption in asymmetric cell division. Biochemical and functional assays

indicated a key role of LRP6 in orchestrating localized Wnt3a signal transduction, including the formation of LRP6/β-catenin/NuMA complexes and local actin cytoskeleton remodeling (Figs. 4 and 5). Considering this, we decided to further investigate the role of LRP6 in mitochondrial apportioning using LRP6KO mESCs undergoing division in contact with Wnt3a-coated beads. Our results show that mitochondria are predominantly distributed equally between daughter cells in LRP6KO mESCs, unlike the asymmetric inheritance observed in wild-type cells (Fig. 7f). These findings highlight the importance of localized Wnt signaling in regulating not only division orientation but also the asymmetric apportioning of mitochondria during cell division, and that these processes are mediated by the LRP6 co-receptor.

## Discussion

The precise orientation of cell division is crucial for epithelial tissues. This orientation determines the position of daughter cells within the

**Fig. 6 | Localized mitotic Wnt3a-sources induce mitochondria polarization.**
**a** Representative transmission electron microscopy images of HeLa cells dividing in contact with Wnt3a-coated beads (upper panel) or BSA-coated beads (lower panel) at two different magnifications. Enlarged views on the right-hand side are boxed in green (top panels) and in orange (bottom panels). Scale bars: 5 μm (left) and 0.5 μm (right). Each cell was divided into circular sectors of 60° and color-coded according to its position relative to the bead (see also Fig. S6a, b). Mitochondrial density was measured in the sector proximal to the bead (shaded in fuchsia) and compared with the opposite sector (colored in green). Identified mitochondria are color-coded with complementary colors (mitochondria are in green for fuchsia sectors, and in fuchsia for green sectors). **b** Quantification of mitochondrial density (number of mitochondria/μm$^2$) and mitochondrial area (percentage of cytosolic area occupied by mitochondria) in circular sectors centered on Wnt3a-beads or BSA coated beads. Mitochondrial density was measured as the ratio between the number of mitochondria in a circular sector proximal to the bead (fuchsia, panel **a**) and a circular sector distal to the bead (green, panel **a**). Mitochondrial area was calculated as the ratio between the fractional area occupied by mitochondria in a circular sector proximal to the bead, and a sector distal to the bead. For each condition, mean and SD are shown for 3 cells, with *$p < 0.05$ by unpaired $t$-test. (Mitochondrial density $p$-value = 0.0258, Mitochondrial area $p$-value = 0.0348). **c** Representative 3D reconstruction of mitochondria networks in metaphase HeLa cells dividing in contact with Wnt3a-beads (left) or BSA-beads (right). Cells were stained with DAPI (blue) to visualize DNA, and Mitotracker CMXRos Red M7512 (white) to visualize mitochondria. Beads are visible by autofluorescence (magenta circle). Scale bar, 5 μm.
**d** Scheme illustrating the quantification protocol used to evaluate mitochondrial density in proximity of the beads (mitochondria in the first two shells 1 + 2, shaded in dark and light green), or distally to the beads (mitochondria in the last two shells 9 + 10, shaded in light and dark pink). Mitochondria are depicted in red, the metaphase plate in blue and the spindle microtubules in gray. **e** Normalized mean fluorescence intensity (MFI) profiles of the mitochondrial signal along the shells for HeLa cells dividing in contact with Wnt3a-beads (blue) or BSA-beads (red). Data are shown as mean ± standard deviation of $n = 31$ for Wnt3a-beads and $n = 19$ for BSA-beads from 3 independent experiments. **f** Violin plots of the enrichment ratio between mitochondria proximal to the beads (shells 1 + 2) and distal to the beads (shells 9 + 10) for the experiments in (**e**). Single dots indicate values of individual cells; black lines indicate median and interquartile range (Wnt3a-beads: Minima = 0.8470147582129417; maxima = 3.215508244042814; center = 1.71112447780187; Q1 = 1.3777520833266055; Q3 = 2.2741812202744316; Q1 percentile = 25; Q3 percentile = 75. BSA-beads: Minima = 1.004871727631172; maxima = 2.370602424904126; center = 1.350212014871593; Q1 = 1.1687770885463094; Q3 = 1.573842442311806; Q1 percentile = 25; Q3 percentile = 75). *$p = 0.027$ by the Mann–Whitney non-parametric $t$-test. **g** Representative 3D reconstruction of mitochondrial networks in metaphase HeLa cells depleted of CENP-F dividing in contact with Wnt3a-beads. ShCtrl expressing cells were used as control. Cells were stained with DAPI (blue) to visualize DNA, and Mitotracker CMXRos Red M7512 (white) to visualize mitochondria. Beads are visible by autofluorescence (magenta circle). Scale bar, 5 μm. **h** Normalized mean fluorescence intensity (MFI) profiles of the mitochondrial signal along the shells for shCtrl (gray) and shCENP-F (gold) HeLa cells dividing in contact with Wnt3a-beads. Data are shown as mean ± standard deviation of $n = 43$ for shCtrl and n = 38 for shCENP-F from 4 independent experiments. **i** Violin plots of the enrichment ratio between mitochondria proximal to the beads (shells 1 + 2) and distal to the beads (shells 9 + 10) for the experiments in (**h**). Single dots indicate values of individual cells; black lines indicate median and interquartile range (Wnt-shCtrl: Minima = 1.039435322606242; maxima 5.546713801743544; center = 1.591726263215826; Q1 = 1.358683020651499; Q3 = 1.9616798567782072; Q1 percentile = 25; Q3 percentile = 75. Wnt-shCENPF: Minima= 0.7904419281729428; maxima = 2.682701499464928; center = 1.4418130782736505; Q1 = 1.1352336757038564; Q3 = 1.6809989229854057; Q1 percentile = 25; Q3 percentile = 75). *$p = 0.036$ by the Mann–Whitney non-parametric $t$-test.

tissue, ultimately influencing its architecture and function. While mitotic spindle positioning is central to this process, how external signals control it remains unclear. The spindle is positioned by microtubule motors, primarily dynein/dynactin complexes, which generate pulling forces on astral microtubules at specific cortical sites. Understanding how extracellular cues direct this intricate process is key to gaining a deeper understanding of tissue development and disease[22]. Here, we show that localized Wnt3a signaling instructs division orientation of epithelial cells orthogonally by recruiting Gαi/LGN/NuMA/dynein/dynactin complexes at the site of Wnt3a signal activation. We further demonstrate a physical interaction between the dynein-activating adaptor NuMA and the activated LRP6 co-receptor, as well as β-catenin.

Using a newly developed proteomic-based protocol, we identified a complex of proteins enriched at the site of mitotic Wnt3a activation. This complex includes not only Wnt pathway components and spindle orientation proteins but also actin remodelers, mitochondrial, and RNA-binding proteins. Consistent with these findings, our imaging studies revealed a striking enrichment of mitochondria near the Wnt3a signal source. Furthermore, in Wnt-mediated asymmetric cell division of mESCs, mitochondria were preferentially enriched by the daughter cell maintaining contact with the Wnt3a source, along with the Wnt effector β-catenin. These findings suggest a broader role for localized Wnt signaling in orchestrating the asymmetric inheritance of cellular components during division.

Our previous studies in cultured mouse embryonic stem cells and human skeletal stem cells have shown that localized Wnt3a signaling can induce asymmetric cell divisions. Here, we use HeLa cells, a human cervical cancer cell line, to demonstrate that localized Wnt3a signaling can direct spindle orientation even in a transformed, non-stem cell context. Using time-lapse imaging, we tracked HeLa cell divisions in contact with a single Wnt3a-coated bead. Remarkably, this localized Wnt signal was sufficient to orient cell division perpendicular to the bead. Furthermore, the mitotic spindle assembled directly orthogonal to the Wnt3a bead, with no further adjustments before metaphase.

This suggests that Wnt3a signaling breaks cortical symmetry and establishes spatial control of spindle pole formation before the cell enters mitosis. The phenomenon of predetermined division orientation has also been observed in *Drosophila* larval neuroblasts. Here, one centrosome remains anchored apically throughout the cell cycle, ensuring directional spindle assembly in subsequent divisions. This apical anchoring depends on Par proteins, key regulators of cell polarity[56]. Taken together, these findings highlight how external cues, whether polarity-driven (as in fly neuroblasts) or Wnt-dependent (as in our study), can break cortical symmetry and pre-determine the orientation of cell division.

Epithelial cells in culture display a striking behavior: they divide parallel to the surface they grow on. This is due to an interplay between the cell adhesion to the surface, mediated by integrins, and the machinery responsible for cell division, the spindle apparatus. Wnt-signaling has been implicated in spindle axis alignment to the substratum in HeLa cells through a mechanism impinging on the Frizzled-2 receptor and a mitotic Plk1 phosphorylation of Dvl2[16]. We analyzed the angle between the mitotic spindle axis and the substratum of cells dividing near Wnt3a-coated beads. Our results indicate that adhesive forces and Wnt3a-dependent microtubule pulling forces act independently on the mitotic spindle. Spindle positioning, therefore, reflects the balance between these two forces. This suggests a model where adhesive forces and localized Wnt signals cooperate to promote divisions perpendicular to Wnt-rich niches. Interestingly, we observed distinct genetic requirements for these two orienting influences. Spindle orientation towards Wnt3a beads required the co-receptor LRP6, β-catenin, Dvl1 and Dvl2. In contrast, alignment to the growth surface relied on LRP6, Axin1, and both Dvl2 and Dvl3. This difference reflects distinct molecular pathways downstream of adhesion and Wnt3a-signaling. However, we cannot rule out the possibility that these differences arise from transcriptional changes induced by the down-regulation of destruction complex components, such as APC or β-catenin, which we used in our analysis. Interestingly, in HeLa cells, APC is dispensable for spindle positioning, while in intestinal crypts, it

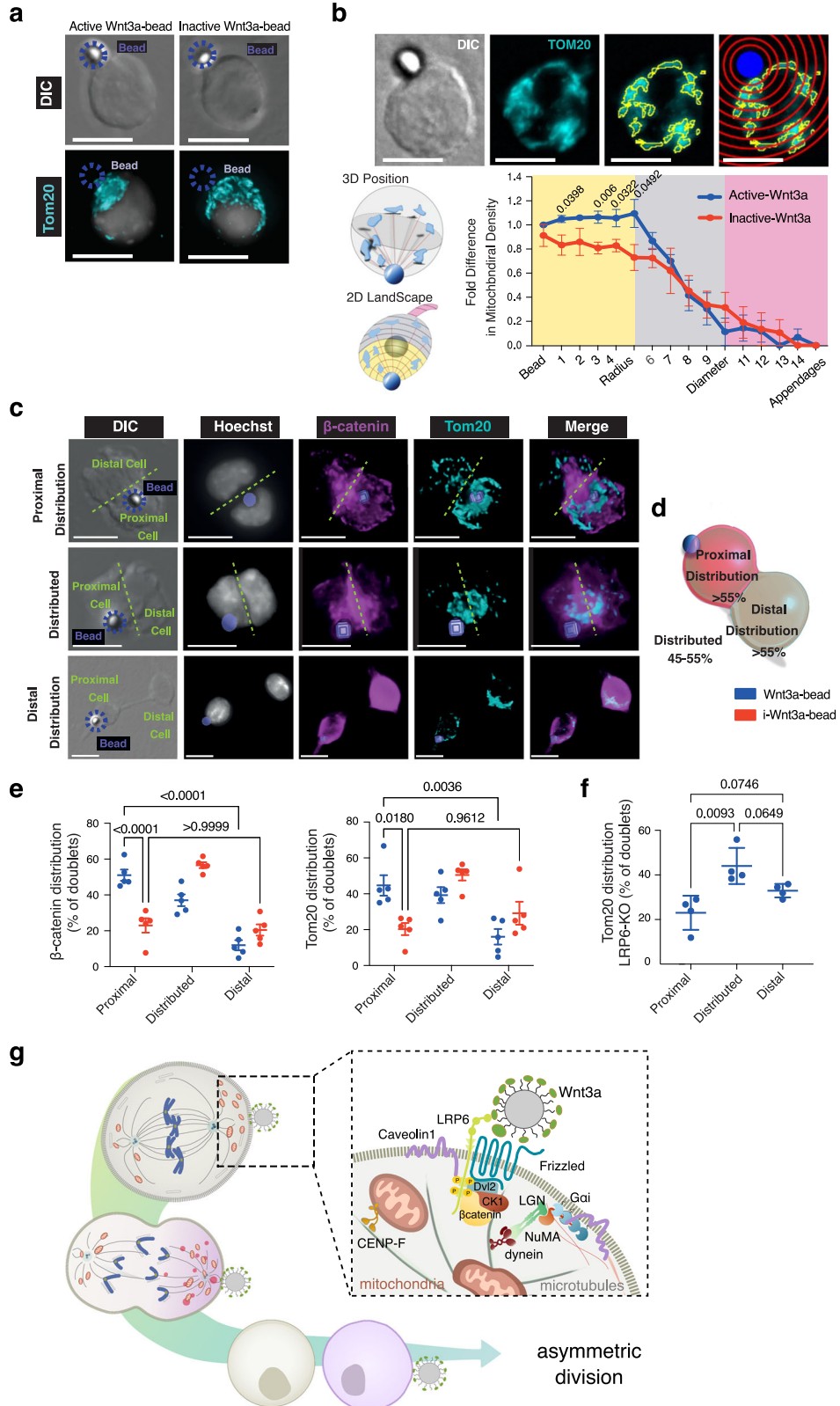

seems to contribute to correct orientation and epithelial integrity programs[57]. In analyzing the molecular requirement for Wnt3-dependent orientation, we did not include analysis of the Frizzled receptors, as we reasoned that these might be cell-specific and would deserve a dedicated study.

To investigate the role of the dynein/NuMA/LGN/Gαi complex in Wnt-mediated spindle orientation, we performed live-cell

imaging of HeLa cells depleted of individual complex components. Depletion of any member of the complex abolished the ability of Wnt3a beads to orient cell division, confirming the necessity of this machinery for Wnt-dependent spindle positioning. We observed a striking accumulation of dynein/NuMA/LGN/Gαi at the contact site between the cell and the Wnt3a bead. This suggests that localized recruitment of this complex generates pulling forces on astral

**Fig. 7 | Localized mitotic Wnt3a-sources instruct the asymmetric apportioning of mitochondria. a** Representative 3D images for oversampling z-stacks, showing single W4 mESCs presented with either active or inactive Wnt3a-beads. DIC images indicate the position of the bead with Tom20 labeled mitochondria and Hoechst-labeled nuclei. The beads are 2.8 μm in diameter. Scale bars: 8.5 μm. **b** Schematic illustrating the covariation of Tom20 reconstructed mitochondrial networks, from a 3D space onto a 2D landscape, using boundary fences set to 1/10th the diameter of the cell. Cell volume-corrected quantification of the mean intensity of Tom20 indicating mitochondrial network density, relative to distance from the bead. Scale: 6 μm. Mean ± SEM, $N \geq 4$ independent experiments, with $n \geq 20$ cells/experiment. The *p*-values < 0.05 are indicated. **c** Representative images of post-division distribution of β-catenin and Tom20-labeled mitochondria within the three categories of distribution. They can be asymmetrically distributed either proximally (top row panels) or distally (bottom row panels) or symmetrically classified as distributed (middle row panels). Scale bars: 9 μm. **d** Schematic of how the different distribution categories in c were quantified. **e** Graphs using the distribution categories as in (**c**) and (**d**), showing that an active Wnt3a-bead (blue circles) increases the percentage of doublets that have ≥55% of both β-catenin and Tom20-labeled mitochondria in W4 wild-type mESCs. Wnt3a-i inactivated beads were used as control (red circles). Mean ± SEM are shown, $N = 5$ independent experiments with $n \geq 18$ cells/experiment. **f** Graph using the distribution categories as in (**c**) and (**d**), showing the impact of Wnt3a-bead on the percentage of doublets that have ≥55% of Tom20-labeled mitochondria in LRP6-KO mESCs. Mean ± SD, $N = 4$ independent experiments $n \geq 18$ cells/experiment. **g** Model of the cortical protein recruitment and spindle orientation mechanism upon localized Wnt3a stimulation in mitotic cells based on our studies.

microtubules, thereby orienting the spindle perpendicular to the Wnt3a source.

Interestingly, while both Gαi1 and Gαi3 isoforms are present in HeLa cells, depletion of either isoform suffices to randomize division orientation. This indicates a lack of functional redundancy between these Gαi isoforms. In contrast to the dynein/NuMA/LGN/Gαi complex, the actin-binding protein Afadin was dispensable for Wnt3a-mediated spindle orientation. Afadin is required for planar divisions in polarized epithelial cells[28]. We think that this difference may reflect the distinct roles of Afadin in polarized versus non-polarized cells[28]. In polarized epithelia, Afadin localizes to adherens junctions, where it contributes to recruiting microtubule motors to control spindle orientation. However, in non-polarized cells like HeLa, Afadin may not be involved in Wnt-dependent spindle positioning.

Mechanistically, oriented asymmetric divisions toward cortical symmetry-breaking cues have been best characterized in invertebrate systems, including *Drosophila* neuroblasts and *C. elegans* zygotes[24]. More recently, artificial symmetry-breaking mechanisms, including photoactivatable targeting of MT motors at the cortex and formation of Par protein caps, have been developed in eukaryotic systems to study the link between polarizing cues, MT cytoskeleton and division orientation[18,23,58]. Less is known about the molecular mechanisms underlying asymmetric partitioning of cell fate determinants driven by localized Wnt3 signals, which remains a long-standing question in the stem cell field. To fill this gap, we developed a proteomic-based protocol to identify cellular components specifically recruited at the Wnt3a-bead contact site. The purification protocol exploits the paramagnetic properties of the beads on which Wnt3a ligands are immobilized, and the high-affinity interaction between the ligands and surface receptors, and was used on a population of metaphase cells enriched by an optimized synchronization procedure. It must be emphasized that this method is applicable to any cell type at diverse cell cycle stages, with ligands other than Wnts. We envision that these key advantages of our method will be harnessed in the future to provide unprecedented molecular information on the cellular response to diverse extracellular signals under physiological and pathological conditions. This will include the identification of the composition of protein complexes transducing extracellular cues and local activation of signaling pathways.

In our study, we analyzed the proteome accumulating at the Wnt3a contact site in metaphase HeLa cells. Consistent with genetic and imaging analyses, we found that effectors of the Wnt pathways are enriched at Wnt3a beads, including the transmembrane Wnt amplifying proteins R-spondin-3 and the surface proteoglycan syndecan-1/4 implicated in Wnt-signaling[40]. The top hit for the proteomic Wnt3a-interactome is the kinase CK1α, poorly characterized in mitosis. By magnetic bead pull-down, we confirmed that CK1α is targeted to the Wnt3a-bead contact site and that its enzymatic activity is essential for Wnt3a-dependent division orientation. Similarly, we proved that membrane-associated caveolins accumulating at the Wnt3a-beads are key players for transducing mitotic Wnt3a signals to the spindle

apparatus, in line with previous reports on the role of these proteins in spindle alignment to adhesive substrates[49]. We showed that Wnt3a-bead-instructed division orientation is triggered by targeting of membrane-associated Gαi subunits at the Wnt3a-contact site, which, by direct interaction, recruit LGN, NuMA and microtubule motors. Gαi1 is found significantly enriched in the Wnt3a-bead interactome, together with NuMA, its cortical interactor 4.1R[59], and with the kinesin Kif2A, recently reported among the mitotic Wnt3a effectors[41]. Further biochemical analysis showed that the entire cortical dynein-anchoring complex, consisting of Gαi/LGN/NuMA/dynactin enriches at the Wnt3a-beads with Dvl2, and that Wnt activation induces the association of the NuMA-containing MT-motors with β-catenin and the co-receptor LRP6. These findings uncovered a direct molecular link between localized mitotic Wnt-signaling and the spindle orientation motors. Collectively, our work extends previous findings on spindle orientation pathways and provides novel insights into Wnt-signaling functions during mitosis.

Our Wnt3a-bead-specific proteomic analysis identified several actin-binding proteins, suggesting a major local rearrangement of the actomyosin cortex and cytoskeleton at the Wnt3a contact site. Among them, we confirmed that the Wnt-dependent F-actin and MT cross-linker MACF1/ACF7 pulls down with the Wnt3a-beads and assists division orientation, somehow mirroring the function described for this protein in Wnt-dependent directional migration of skin stem cells[51]. Intriguingly, MACF1 might also be implicated in mitochondrial transport toward the Wnt source, as reported in skeletal muscle cells[60]. In addition, Wnt3a-bead proteomic and pulldown experiments showed that myosin1B and myosin1C are recruited to the Wnt3a sites. These myosin-1 family members are known to link the actin cytoskeleton to cellular membranes, promoting membrane remodeling and regulation of actin dynamics[61]. By TIRF microscopy, we showed that myosin1C contributes to stabilizing actin-mediated mitotic cell attachment to Wnt3a in a LRP6-dependent manner. These findings support the notion that local anchorage to the substrate of the mitotic cells increases at the Wnt3a contact site in order to sustain traction forces on astral MTs required for attaining division orientation orthogonal to Wnt3a, adding an extra layer to our understanding of mechanisms of division orientation instructed by localized extracellular cues presented directionally to mitotic cells.

Our study reveals a connection between localized Wnt signaling and mitochondrial dynamics. We found that mitotic cells exhibit a remarkable enrichment of mitochondria near sites of Wnt3a activation. This asymmetric mitochondrial distribution was evident in both HeLa cells and mESCs exposed to Wnt3a-coated beads, suggesting a conserved mechanism across diverse cell types. This finding was corroborated by multiple lines of evidence. Proteomic analysis revealed a prominent enrichment at Wnt3a-beads of mitochondrial proteins and transport factors, such as myosin-19[53,54], the microtubule-binding protein CENP-F[55] and the outer mitochondrial membrane GTPase Miro-2[62], which have been implicated in mitochondrial transport. Electron microscopy confirmed the asymmetric distribution of mitochondria in

metaphase cells, with increased mitochondrial density observed at Wnt3a contact sites. Mechanistically, we show that in HeLa cells, ablation of CENP-F abolishes mitochondrial polarization to Wnt3a-beads, indicating that it promotes mitochondrial transport toward the site of Wnt signaling by anterograde movement on microtubules.

Importantly, this Wnt-mediated mitochondrial polarization was linked to asymmetric cell fate in mESCs. Daughter cells proximal to the Wnt3a source, which predominantly adopted an embryonic stem cell fate, inherited a greater proportion of mitochondria compared to their distal counterparts, which were biased towards epiblast stem cell differentiation. Our previous data have shown that deletion of LRP6 abrogates segregation of cell fate choice markers and thus asymmetric cell division[20]. Here, we show that LRP6 deletion also disrupts the asymmetric partitioning of mitochondria during cell division in the presence of localized Wnt3a. This intriguing observation raises the possibility that localized mitochondrial enrichment may directly influence cell fate decisions, a question we are actively pursuing in ongoing studies. Overall, our work unveils a novel aspect of Wnt signaling, demonstrating its capacity to spatially organize mitochondria within cells. This finding has broad implications, suggesting a potential mechanism by which localized extracellular cues could shape intracellular organization and influence cell fate. Furthermore, the conserved nature of this phenomenon, observed in both stem cells and cancer cells, highlights its potential relevance for understanding both development and disease.

Finally, our proteomic analysis yielded an intriguing observation: a striking enrichment of RNA-binding proteins at cortical sites of Wnt3a activation. These proteins are involved in a diverse array of RNA-related processes, including RNA modification, splicing, and transport. This finding suggests that localized Wnt signaling may exert a profound influence on RNA fate and function. This observation is particularly compelling given the precedent for asymmetrically inherited mRNA molecules acting as determinants of cell fate. For example, in dividing *Drosophila* neuroblasts, the unequal partitioning of specific mRNAs has been shown to specify distinct daughter cell identities. While our study focused on identifying the protein players involved in Wnt-mediated spatial control, future investigations will delve into the functional consequences of this localized enrichment of RNA-binding proteins. Specifically, we aim to determine whether Wnt signaling leads to the asymmetric inheritance of specific mRNA molecules and, if so, how this might contribute to divergent transcriptional programs and cell fate decisions in daughter cells. Investigating these questions in stem cell systems, where Wnt signaling critically regulates self-renewal and differentiation, could provide especially valuable insights.

In conclusion, our work highlights the power of Wnt3a artificial niches coupled to proteomics and imaging to address the principles of asymmetric divisions and stem cell self-renewal. Our data support a model in which localized Wnt3a activation couples perpendicular division orientation relative to the niche with the asymmetric recruitment of key molecules. These molecules include spindle motors, myosins-1, and Wnt effectors, which are able to remodel and regulate locally the cytoskeleton, promoting mitochondrial polarization and asymmetric apportioning. These mechanisms are coupled to asymmetric cell division and cell fate decisions (Fig. 7g). In addition, we provided a general approach to study asymmetric divisions in different stem cell systems and conditions, possibly uncovering molecular pathways important for tissue homeostasis or altered in diseases.

## Methods

### Cell Culture and drug treatments

HeLa cells (ATCC, CCL-2) and HEK293T cells (ATCC, CRL-11268) were cultured at 37 °C in a 5% $CO_2$ atmosphere, in Dulbecco's Modified Eagle Medium (DMEM) supplemented with 10% FBS, 1% L-glutamine and 50 μg/ml penicillin/streptomycin. For HeLa cells infected with pLKO.1

lentivirus, the medium was supplemented also with 1 μg/ml puromycin or 0.2 μg/ml hygromycin. To depolymerize astral MTs, HeLa cells were treated with 20 nM nocodazole (Sigma Aldrich) for 4 h after thymidine release. Wnt3a conditioned medium was prepared from mouse L-cells stably expressing and secreting mouse Wnt3a (ATCC, CRL-2647), as described[47]. DKK1-conditioned medium was obtained with transient transfection with PEI of HEK 293T cells with a retrovirus p-BaBe (generous gift from Prof. Myriam Alcalay, IEO). For spindle orientation analyses of Fig. 1e and S1n–p and immunofluorescence analyses of Fig. 2b and e, HeLa cells were plated on fibronectin-coated coverslips (5 μg/ml, Roche) and pre-synchronized by a single thymidine (2.5 mM, Sigma T1895) block/release. For time-lapse filming of Fig. 1b and analogous spindle orientation analyses, cells were synchronized by a single thymidine pulse. The protocol was further adjusted to add Wnt3a-beads and, when needed, label DNA by SirDNA (200 nM, Tebu Bio SC007). For co-immunoprecipitation experiments of Fig. 4b and c, HeLa cells were synchronized in G2/M using 9 μM RO-3306 (Merck Life Science, SML0569) for 16 h, released for 30 min, subsequently supplemented with 10 μM MG132 (Bio-techne) for 1 h to inhibit the proteasome, and then harvested by shake-off. For Wnt-dependent transcriptional activation (Fig. S2), after thymidine release, HeLa cells were treated with 10 μM CHIR-99021 (Selleckem Catalog No. S2924) for 4 h, the drug was washed out, and Wnt3a-beads were seeded with cells as described below. For the inactivation of Wnt target gene transcription, HeLa cells were treated with 10 μM LF3 (Selleckchem, Catalog No. S8474), Wnt3a-beads were seeded with cells after 20 h of treatment and filmed as described below.

### Lentiviral transduction and transfection

To deplete the expression of spindle orientation proteins and Wnt components, the shRNA sequences targeting the specific genes (listed in Supplementary Table 2) were cloned into pLKO.1 lentiviral vectors (Addgene #8453) carrying a puromycin or a hygromycin reporter, and used to generate stably interfered HeLa cell lines by single infection. Protein depletion was monitored by western blot or RT-qPCR for APC, Dvl1, Gαi and caveolin-1/2. For IP experiments and cortical NuMA rescue experiments (Fig. 4), human LRP6 was amplified from the pCS2-LRP6 vector (Addgene #27282) and cloned into a pEGFP-N1 vector. A sh-resistant EGFP-tagged LRP6 construct was generated by targeted mutagenesis, introducing four silent base substitutions in the region targeted by the sh-LRP6-A hairpin. To obtain a non-phosphorylatable LRP6 mutant (named LRP6-M5), residues Ser1490 (M1), Thr1529 and Thr1530 (M2), Thr1572 (M3), Ser1590 (M4) and Ser1607 (M5) were replaced with alanine by sequential QuikChange mutagenesis. All constructs were sequence verified. LRP6 constructs were transfected into HeLa cells by Lipofectamine-3000 (Life Technologies) according to the manufacturer's instructions.

To generate HeLa cells stably and homogeneously expressing nearly endogenous levels of LRP6-APEX2, HeLa cells were transfected with pCS2 LRP6-APEX2-Ires-eGFP-PAC (Addgene #180142), and clonally expanded after puromycin selection. The clone used in Fig. 4b was chosen by Western Blot analysis. For TIRF microscopy experiments, a myosin1C-YFP containing plasmid (generous gift from the Matthew J. Tyska laboratory, Vanderbilt University) was transfected into HeLa cells stably expressing shLRP6 or shCtrl using Lipofectamine 3000 (Invitrogen) according to the manufacturer's instructions. For mitochondrial quantization, HeLa cells were infected with pHR GFP-CaaX (Addgene #113020, generous gift from Prof. Scita, IFOM Milan) and sorted on a BD-FACSDiva 8.0.2.

### RT-PCR analysis

Total RNA was extracted from HeLa cells using the Qiagen RNeasy Mini kit with DNase I on-column treatment. 1 μg of RNA was retro-transcribed using the LunaScript RT SuperMix Kit (New England BioLabs, E3010G) according to the manufacturer's instructions. Real-time

quantitative PCR (RT-qPCR) was performed by amplification of 10 ng of cDNA using Luna Universal qPCR Master Mix (New England BioLabs, M3003G) following the manufacturer's instructions, using the primers used are listed in Supplementary Table 2.

## Wnt3a-Beads preparation

Preparation for Wnt3a-coated microbeads was done following published protocols[18]. Briefly, 2.8 µM magnetic Dynabeads M-270 Carboxylic Acid (Invitrogen) were activated by 30 min treatment with 50 mg/ml N-(3-dimethylaminopropyl)-N-ethylcarbodiimide hydrochloride EDC) and 50 mg/ml N-Hydroxysuccinimide in 25 mM MES buffer, pH 5.0. Activated beads were incubated with pure murine Wnt3a (PeproTech, 315-20) at 40 µg/ml for 1 h at room temperature, or with the same concentration of BSA (Life Technologies, AM2616). To inactivate immobilized Wnt3a-ligands, Wnt3a-beads were incubated in 100 µl of 20 mM DTT for 30 min at 37 °C with slow vertical rotation.

## TOP-FLASH assay

For the TOP-Flash assay of Fig. S1a, b, HEK 293T TCF7-Luciferase cells were seeded with different concentrations of Wnt3a-CM, Wnt3a-beads or BSA-beads for 24 h. After lysis, the luciferase activity was measured using the Luciferase Assay System kit (Promega, E1500) following the manufacturer's instructions. The luminescence was read using Glomax Discovery. The luminescence measured was normalized by the cell concentration.

## Microscopy

Confocal images were acquired on a Leica SP8 AOBS confocal microscope controlled by LasX Leica confocal software. For HeLa cells analysis of Fig. 2b, d, e, a ×63 oil-immersion objective lens (HCX Plan-Apochromat ×63 NA 1.4 Ldb Bl) or Nikon Spinning Disk CSU-W1 with NIS-Elements AR software, a Plan Apo lambda ×100/1.45 NA oil immersion objective lens was used. Images shown in Figs. 1f and S1n were acquired on a Leica SP8 confocal microscope controlled by the LasX Leica confocal software with a ×63 oil-immersion objective lens (HC PL Apochromat ×63/1.4NA CS2). For live-imaging time-lapse microscopy, we used a Nikon Eclipse Ti microscope equipped with a Lumencor SpectraX fluorescence illuminator, an Andor Zyla 4.2 sCMOS camera, a multiband dichroic mirror, single-band emission filters and a PlanApo ×20/0.75NA objective lens (Nikon Europe B.V.). All images were processed using the software Fiji40[63].

## TIRF microscopy and analysis of spreading assay

MATTEK 35 Mm Dish I No. 1.5 Coverslips I 10 Mm G were first functionalized with Wnt3a and then filmed for TIRF microscopy. Wnt3a was deposited at 50 µg/µl concentration on the coverslip, and it was absorbed on the surface for 1 h at 37 °C. Cells were synchronized using 9 µM RO3306 for 16–20 h. After the release from RO3306, cells were shaken off the plate and dropped on the Mattek for the spreading assay. For image acquisition, a Leica HC PL APO CS2 ×63/1.40 oil immersion objective was used. DIC and total internal reflection fluorescence (TIRF) microscopy of live time-lapse of spreading cells was performed on a Leica AM TIRF MC system at the IFOM Imaging Facility. A 488 nm laser was used for fluorochrome excitation. Environmental conditions were maintained at 37 °C and 5% $CO_2$ using an Okolab temperature (H201-T module) and $CO_2$ (DGTCO2BX module) control system.

The spreading assay was performed on the Mattek dish in serum-free DMEM w/o phenol red (Euroclone, ECB7504L), cells were then followed by acquiring the YFP-myosin1C signal for 40 min at a frame rate of 0.1 fps. Image analysis was then performed with Fiji[63]. In Fig. 5g and h, the portion of the cell in adhesion with the surface was detected by identifying the presence of myosin signal in time and by calculating the area of Myosin1C-positive regions. In particular, myosin-positive pixels were identified by setting a threshold using a Moments

algorithm on the latest time frame. For every time point, the average myosin-positive area and its standard deviation were calculated on 11 cells in 3 independent replicas. In Fig. 5i–l, the persistence of the Myosin1C signal in time was calculated. This analysis was performed by initially identifying the presence/absence of myosin signal on a pixel basis (same as above). Then, a time-stack projection was built, where every pixel value represents the time for which the myosin signal was positively detected (Fig. 5i). Finally, the area of myosin persisting for at least 22.5 min (around half of the time lapse duration) was calculated for 11 cells in 3 independent replicas (Fig. 5l).

## Immunofluorescence

For immunofluorescence, HeLa cells were plated on 13 mm coverslips coated with 5 µg/ml fibronectin. Cells were fixed according to the different antibodies with methanol at −20 °C for 10 min or with 4% paraformaldehyde (PFA) at room temperature. The PFA-fixed samples were permeabilized with 0.3% Triton X-100 in PBS for 5 min. For all conditions, blocking was performed with 3% BSA in PBS for 1 h at room temperature. Cells were stained with the primary antibodies (see Supplementary Table 3 for antibody dilutions), in a solution of 3% BSA + 0.05% Tween-20 for 2 h at room temperature within a humidified chamber. For p150 staining, HeLa cells were fixed with methanol at −20 °C for 10 min, followed by permeabilization with 0.1% Triton X-100 in PBS for 10 min. Blocking was performed with 3% BSA in 0.1% Triton X-100 in PBS for 1 h at room temperature, and cells were stained with primary antibodies overnight at 4 °C. Incubation of secondary antibody anti-mouse or anti-rabbit AlexaFluor-647 (1:400, ThermoFisher Scientific #A32787 and #A32795), or anti-mouse AlexaFluor-488 (1:400, ThermoFisher Scientific #A32766) or anti-mouse AlexaFluor-555 (1:400, ThermoFisher Scientific #A32773), was carried out for 1 h at room temperature, in a dark, humidified chamber. DNA was stained with DAPI (Merck, 32670).

## Live cell imaging

For a live cell imaging experiment with the Wnt3a-beads, HeLa H2B-GFP or HeLa shRNA cells treated with SirDNA (Cytoskeleton, sc007) were seeded in an 8-chamber Ibidi (Twin Helix Srl), using 10.000 cells/cm² for each chamber. HeLa cells were thymidine-synchronized and, after release, Wnt3a or BSA-coated beads were seeded with cells at a concentration of 0.01 µg/µl for about 4 h before the expected mitotic peak. For time-lapse videorecording, images were acquired for 12 h every 5 min using a fully motorized Nikon Eclipse Ti microscope controlled by the NIS-Elements AR software (v. 5.11.01). Differential interference contrast (DIC) and far-red or the GFP channels were acquired to visualize the cells and the SirDNA or the GFP signals, respectively. For Fig. 1d, metaphase frames from time-lapse microscopy of HeLa cells stably expressing GFP-H2B were used to calculate the percentage of spindles pre-oriented versus re-oriented toward the Wnt3a-beads. For each frame, the GFP-H2B signal is color-coded differently, with the position of the metaphase plate marked by a dashed line. The re-orientation extent was calculated considering the frequency at which the metaphase plate rotates more than 20° between two consecutive frames. T-test was performed for two independent experiments with about 10 cells per condition.

## Spindle orientation measurements

Three mitotic spindle orientation angles were examined in HeLa cells: $\alpha$ and $\gamma$ angles were measured in from $X$–$Z$ confocal sections, while the angle $\beta$ was measured from time-lapse frames acquired in the $X$–$Y$ plane, as detailed below. Quantification of $\alpha$ angle between the spindle axis and the substrate (Figs. 1f, S1n–p): HeLa cells were plated on fibronectin-coated coverslips and stained with γ-tubulin or NuMA to visualize the spindle poles and DAPI to visualize the DNA. Cells were imaged in $X$–$Z$ optical sections passing through the spindle poles. To determine the orientation of metaphase spindles, the angle formed by

a line passing through the spindle poles and the substratum was measured, exploiting the angle tool of the software Fiji. Quantification of $\beta$ angle between spindle axis and bead in the $X$–$Y$ planes (Figs. 1b, c, d–g, 2a–d and 5b–d): the $\beta$ angle was measured as the acute angle between the metaphase plate and the line passing through its center toward the bead in the $X$–$Y$ plane in the last metaphase frame of time lapse filming. The distribution of measured $\beta$ angles was graphed in a rose plot from 0° to 90° using Oriana software (Kovach Computing Services). Quantification of $\gamma$ angle between the spindle axis and the bead in the $X$–$Z$ plane (Fig. 1f): the $\gamma$ angle was measured as the angle between the line passing through the spindle poles and the line from one distal centrosome to the bead. Statistical analysis of angle distributions was performed in Prism with the Kruskal–Wallis test.

### Quantification of NuMA complex cortical/cytosol ratio at the Wnt3a-beads and BSA-beads

HeLa cells in contact with Wnt3a or BSA-coated beads were thymidine-synchronized, fixed and stained for NuMA, LGN, Gαi1 or the dynactin subunit p150Glue. To quantify NuMA, LGN, Gαi1 and p150Glue cortical/cytosol ratio signal at bead contact site (Fig. 2c), confocal sections of metaphase cells were analyzed in Fiji[63] with the following pipeline. A 10-pixel-wide line (600 nm wide) was drawn from the bead contact site to the spindle pole, and the signal intensity profile along the line was quantified. Note that each intensity value was obtained by integrating intensities over the line width. The amount of protein at the cortex was calculated by integrating for 10 pixels the signal intensity along the line from the bead contact site (towards the cell center). At the same time, the amount of the protein in the cytosol was calculated by integrating for 10 pixels the signal intensity along the line from 30 pixels apart from the bead contact site (towards the cell center). This pipeline is graphically shown in Fig. S3c, d. The cortical/cytosol signal intensity ratio was then quantified for both Wnt3a- and BSA-coated beads. T-test was performed for 45 cells in three independent experiments.

To quantify astral microtubules at the Wnt3a/BSA-bead contact site (Fig. 2e), a Nikon Spinning Disk CSU-W1 with NIS-Elements AR software, a Plan Apo lambda 100x/1.45 NA oil immersion objective lens z-projection images of metaphase HeLa cells in contact with Wnt3a- or BSA-bead stained for α-tubulin were used. The astral microtubule intensity was quantified within a 10 µm diameter region centered on the bead, excluding the region outside the cell.

### Immunoprecipitation experiments

For the immunoprecipitation of LRP6-FLAG of Fig. 4b, HeLa cells clone expressing pCS2 LRP6-FLAG-APEX2 were treated with Wnt3a or DKK1-conditioned media at 1/3 of the total culture medium for 3 h, synchronized in metaphase and harvested by shake off. Mitotic cells were lysed (50 mM Na–HEPES pH 7.5, 2 mM EGTA, 2 mM MgCl₂, 150 mM KCl, 10% glycerol, 0.2% NP-40, protease inhibitor cocktail) for 30 min, and then the lysates were cleared by centrifugation at 13,000×g for 30 min. Five hundred micrograms of cell extract were incubated with 20 µl slurry of anti-FLAG M2 affinity gel beads (Merck, A2220) for 4 h with gentle agitation on a wheel at 4 °C; as a negative control, the parental HeLa WT mitotic lysate was used. After supernatant removal, beads were washed 3 times with 200 µl of wash buffer (50 mM Na–HEPES pH 7.5, 2 mM EGTA, 2 mM MgCl₂, 150 mM KCl, 10% glycerol, 0.1% NP-40), and 6X Laemmli sample buffer was added to the beads for SDS–PAGE and immunoblotting analysis.

For the immunoprecipitation of LRP6-GFP of Fig. 4c, HeLa cells were transfected with pEGFP-N1-LRP6-WT or M5, treated with Wnt3a-conditioned media as previously described, synchronized in metaphase and harvested by shake off 48 h post-transfection. Mitotic cells were lysed (50 mM Na–HEPES pH 7.5, 2 mM EGTA, 2 mM MgCl₂, 150 mM KCl, 10% glycerol, 0.2% NP-40, protease inhibitor cocktail) for 30 min, and then lysates were cleared by centrifugation at 13,000×g for 30 min. Five hundred micrograms of cell extract were incubated

with 20 µl slurry of anti-GFP mAb-agarose beads (MBL, D153-8) for 2 h with gentle agitation on a wheel at 4 °C; as a negative control, the parental HeLa WT mitotic lysate was used. After supernatant removal, beads were washed 3 times with 150 µl of wash buffer (50 mM Na–HEPES pH 7.5, 2 mM EGTA, 2 mM MgCl₂, 100 mM KCl, 10% glycerol, 0.075% NP-40), and 6X Laemmli sample buffer was added to the beads for SDS–PAGE and immunoblotting analysis.

### Magnetic Wnt3a-beads pull-down

HeLa cells were synchronized in metaphase by RO-3306 and MG132 treatment as described above, adding Wnt3a- or BSA-coated beads concomitantly to the MG132 treatment. Per each pull-down, around 500 µg were obtained at a concentration between 5 and 10 µg/µl. To preserve cytosolic constituents adsorbed on the microbeads, cells were harvested by shake-off and lysed in 60 µl of a mild lysis buffer composed of 75 mM Na–HEPES pH 7.5, 1.5 mM EGTA, 1.5 mM MgCl₂, 0.15 M KCl, 15% glycerol, 0.2% NP-40, protease inhibitor cocktail, 1 µl DNA Benzonase Nuclease 25 kU. Mitotic Wnt3a effector complexes were isolated from the total lysate by taking advantage of the magnetic properties of the beads, without lysate clearance. BSA-coated beds were used as a control. After supernatant removal, beads were washed 3 times with 0.2 ml of lysis buffer, and Laemmli 6X sample buffer was added to the beads for SDS–PAGE and immunoblotting analysis. Samples for proteomic analyses were generated similarly, resuspending the beads in PBS after the washing steps.

### Mass spectrometry

For the proteomic analysis of Fig. 3, proteins isolated by magnetic pull-down in triplicate were separated by SDS-PAGE using 4–12% NuPAGE Novex Bis–Tris gels (Invitrogen) and stained by Coomassie-Brilliant Blue (Abcam). Bands from gel were excised and subjected to in-gel protein digestion. Gel pieces were reduced for 1 h at 56 °C with reduction buffer (10 mM DTT in 50 mM AmBic), and then alkylated in the dark for 45 min with alkylation buffer (55 mM iodoacetamide and 50 mM AmBic). Trypsin enzymatic digestion was performed overnight with 100 ng/µl trypsin (Promega) in 50 mM Ambic at 37 °C. Peptides were desalted and concentrated with C18 Stagetips (Proxeon Biosystems). Finally, eluted peptides were vacuum-dried (Eppendorf concentrator 5301), resuspended in 0.1% formic and injected on a Q-Exactive™ Plus Hybrid Quadrupole-Orbitrap™ Mass Spectrometer (Thermo Fisher Scientific). A linear gradient ramping from 5% to 25% of Buffer-B (80% ACN and 0.1% FA) in 60 min was applied with a flow of 250 nL/min. Full scan MS spectra were acquired in a range of $m/z$ 375–1650.

### LC–MS/MS analysis, proteins identification and quantitation

Raw data files were analyzed using the peptide search engine Andromeda integrated into the MaxQuant software suite (version 2.1.4.0)[64], with the following parameters: human protein database, Oxidation (M), Acetyl (Protein N-term), as variable modifications, peptide false discovery rate (FDR) 0.01, maximum peptide posterior error probability (PEP) 1, protein FDR 0.05, minimum peptides 2, at least 1 unique, minimum length peptide 6 amino acids. Statistical analyses were performed using the Perseus software[65] and R (version 4.3.1). Significance between three replicates was assessed by Student's T-test, considering a threshold $p$-value < 0.05 and Log2 fold change > 1. Data were visualized in GraphPad and R (Fig. 3b). Gene Ontology (GO) analysis of Fig. 3d and e was performed using ClusterProfiler and plotted in R[66]. Protein-protein interaction networks were generated in Cytoscape v. 3.9.1[67].

### Immunoblotting

For western blot analysis, cells were collected and lysed in lysis buffer containing 75 mM HEPES pH 7.5, 1.5 mM, EGTA, 1.5 mM MgCl₂, 150 mM KCl, 0.1% NP40 and 15% glycerol and protease inhibitors (Calbiochem,

539134). 20 µg of cell lysates were resolved by SDS-electrophoresis and transferred to a nitrocellulose membrane. Blocking was performed in TBS containing 0.1% Tween-20 and 5% low-fat milk. Primary antibody incubation was performed at 4 °C for 16 h (see Supplementary Table 3 for antibody dilution). Secondary antibody was incubated for 2 h at room temperature.

### Electron microscopy

Transmission electron microscopy was performed at the ALEMBIC Facility (San Raffaele Hospital, Milan). Cells were fixed with 2.5% glutaraldehyde in 0.1 M cacodylate buffer, pH 7.3, for 1 h and postfixed with reduced osmium (1% $OsO_4$, 1.5% potassium ferrocyanide in 0.1 M cacodylate buffer, pH 7.4) for 1 h on ice. After several washes in milli-Q water, samples were stained en bloc with 0.5% millipore-filtered uranyl acetate. Finally, samples were rinsed in $dH_2O$, dehydrated with increasing concentrations of ethanol, embedded in Epon resin and cured in an oven at 60 °C for 48 h. Ultrathin sections (70 nm) were obtained using an ultramicrotome (UC7, Leica microsystem, Vienna, Austria), stained with uranyl acetate and Sato's lead solutions. After staining, sections were observed using the transmission electron microscope Talos L120C (FEI, Thermo Fisher Scientific). Images were acquired with a CETA 4 × 4k CMOS camera (Thermo Fisher Scientific).

### Mitochondria quantification in HeLa cells from EM images

Microscopy Image Browser was used to segment and analyze mitochondrial distribution[68]. Cells were divided into circular sections of 30° each. Two slices centered on the beads were considered for the circular sector proximal to the bead, and another two on the opposite side for the circular sector distal to the bead. We decided to use circular sections to analyze mitochondrial density, giving more weight to the cytoplasmic area close to the membrane than to the one close to the metaphase plate, where mitochondria are fewer. Mitochondrial density was measured as the ratio between the number of mitochondria in a circular sector proximal to the bead and a circular sector distal to the bead. Mitochondrial area was calculated as the ratio between the fractional area occupied by mitochondria in a circular sector proximal to the bead and a sector distal to the bead. Mean and SD are shown for three cells, with *$p < 0.05$ by unpaired t-test.

### Mitochondria quantification in HeLa cells dividing in contact with Wnt3a/BSA-beads

To evaluate mitochondrial distribution in HeLa cells dividing in contact with beads, cells were stained with Mitotracker (CMXRos Red M7512), DAPI and L1CAM (or cells stably expressing GFP-CAAX were used). Cells were imaged with a spinning disk X-Light V3 module (CrestOptics S.p.A.) mounted on an Eclipse Ti2 fluorescence microscope (Nikon Europe B.V.) equipped with solid-state lasers (Lumencor Celesta light engine), a sCMOS camera (Kinetix, Teledyne Photometrics) and a ×100/1.49 NA oil immersion objective lens (Nikon Europe B.V.). Overall, four channels and 127 Z planes spaced by 0.2 µm were acquired. Images were deconvolved using the Blind algorithm method within the deconvolution module of NIS-Elements software (Nikon Europe B.V.). The deconvolved images were analyzed with a custom Python script. For each field of view, single cells were segmented using DAPI and L1CAM/CaaX-GFP channels with the Cellpose 3D cyto3 model[69,70]. Bead particles were segmented from the bead channel using Yen's thresholding method[71]. Mitotic cells were paired with the closest bead particle based on the Euclidean distance between their centroids. The cell diameter was divided into 10 equal increments, and concentric spherical masks were generated around the bead centroid. Each shell mask was obtained by subtracting the previous (inner) sphere mask, resulting in distinct annular regions. Mean fluorescence intensity (MFI) of the mitochondrial channel was computed for each shell and normalized to the mean intensity of the whole cell. The normalized intensity profiles were used to define proximal (shells 1–2) and distal (shells 9–10) intervals. For each cell, the integrals of the MFI profiles for these intervals were calculated, and their ratio (proximal/distal mitochondrial enrichment ratio) was determined.

### Wnt-mediated asymmetric cell division of mESCs

Wild-type W4 (129S6/SvEvTac) or LRP6 knock-out (LRP6 KO)[20]. mESCs were maintained in ESC basal media containing Advanced DMEM/F12 (cat. no. 12634028, ThermoFisher), 10% ESC-qualified fetal bovine serum (FBS, cat. no. ES-009-B, Millipore), 1% penicillin–streptomycin (cat. no. P4333, Sigma), 2 mM GlutaMax (cat. no. 35050061, ThermoFisher), 50 µM β-mercaptoethanol (cat. num. 21985-023, ThermoFisher) and 1000 U/ml recombinant Leukemia Inhibitory Factor (LIF; cat. no. 130-095-775, Miltenyi), supplemented with with 2i: MEK inhibitor PD0325901 (1 µM, cat. no. 130-104-170, Miltenyi) and GSK3 inhibitor CHIR99021 (3 µM, cat. no. 130-104-172, Miltenyi). To passage ESCs, colonies were rinsed with PBS, treated with 0.25% trypsin-EDTA (T-E, cat. no. 25200056, ThermoFisher) for 4 min at 37 °C, 5% $CO_2$, resuspended using ESC basal media and centrifuged at 1200×$g$ for 4 min. Pelleted cells were resuspended in ESC basal media, counted and seeded in a clean tissue culture-treated six-well plate at low density (~7000 cells/well) in complete ESC media (containing LIF and 2i). 24 h before asymmetric cell division assay, 2i was withdrawn and replaced with soluble Wnt3a protein (200 ng/ml).

For fixed cell analysis, cells were previously counted in the amount of 3500 cells per well and mixed together with either active, 0.35 µg, or inactivated Wnt3a beads, and plated in a volume of 40 µl onto a single well of an 8-well glass slide (ICN Biomedicals), previously coated with 0.1% human fibronectin (Corner). The slides with the cells were incubated for either 7 or 15 h at 37 °C in the absence of 2i in the appropriate media. Cells were then fixed with 4% paraformaldehyde (PFA) (Sigma) in PBS for 8 min at room temperature, and washed three times with staining buffer, which permeablise the cells. The samples were blocked in 10% donkey serum for 1 h at room temperature (Sigma), before adding the appropriate primary antibodies, Tom 20 (sc-17764, 1:500) and β-catenin (BD610153, 1:500) and incubating overnight at 4 °C. The wells were then washed three times with staining buffer before the secondary antibody was added. Cells were incubated with secondary antibody for 1 h at room temperature, washed 3 times in staining buffer, and then mounted in ProLong™ Gold Antifade Mountant with DAPI (ThermoFisher, Catalog no. P36931). Fixed cells were imaged with a Zeiss inverted Axio Imager epifluorescence microscope, equipped with a CoolSNAP HQ2 CCD camera, at ×40/1.3 or ×63 /1.4 oil-based objective. Cells were selected as follows: Singlets/Doublets with a minimal number of beads that have aggregated and are only present at one side of the cells and therefore can induce polarity, or with no beads as a comparable control. An over-sampling technique was used with Z-stacker at 0.2 µm intervals to acquire the images taking at ×40 or ×63, which were then deconvolved in Volocity software. Cell and bead structures were reconstructed in 3D using Volocity software. For singlet polarization analysis, the distance of the detected mitochondrial networks from the bead was measured. This data was then transferred to MATLAB for further analysis. To normalize the distance measurements to the size of the cell, the cell volume was approximated as a sphere. Distances were segmented into intervals, or "fences," each representing one-tenth of the spherical diameter. This conversion allowed for the mapping of the 3D positions of the mitochondrial networks onto a 2D landscape. The mean intensity of Tom20 within the detected mitochondrial networks, representing mitochondrial density, was then analyzed in relation to the bead distance, based on the fence in which each network was first detected.

The distribution of doublets for both β-catenin and Tom20 was analyzed by drawing a dividing line between the proximal and distal regions using the nuclei and β-catenin stains as guides. The doublets were then categorized based on the distribution of β-catenin and

Tom20. If more than 55% of the intensity was localized in either the proximal or distal cell, the doublet was characterized accordingly. Otherwise, if the intensity was between 45% and 55% in both regions, it was classified as distributed. Refer to Fig. 7d for further details. This gives a population analysis displayed as a percentage of doublets for each distribution.

## Statistical analysis

Statistical analyses were performed using GraphPad Prism (GraphPad Software, San Diego, CA). Details of each statistical test are indicated in the figure legends: statistical differences between the 2 groups were analyzed using the unpaired Student $t$ test (Figs. 2c–e, 5c, d), or the Mann–Whitney $U$ nonparametric test (Fig. 6f). For analyses including more than 2 groups (Figs. 1d–h, 2a, 5b) the Kruskal–Wallis ANOVA non-parametric test was applied. The number of independent experiments and the number of cells of each analysis are available in the figure legends. In each presented scatter plot with a bar, the internal bar represents the mean of the distribution with the corresponding standard deviation. Significance thresholds were defined as $p$-value = 0.05.

## Figure preparation

All schemes and cartoon models presented in the manuscript have been prepared with Affinity Designer (Serif Europe) or Adobe Illustrator.

## Reporting summary

Further information on research design is available in the Nature Portfolio Reporting Summary linked to this article.

## Data availability

The mass spectrometry proteomics data have been deposited to the ProteomeXchange Consortium via the PRIDE partner repository with the dataset identifier PXD069774. Other data supporting the findings of this manuscript are available from the corresponding authors upon request. Source data are provided with this paper.

## Code availability

The Python script used to quantify the mitochondrial distribution in HeLa cells dividing in contact with Wnt3a/BSA-beads will be made available upon request.

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

## Acknowledgements

We thank the scientists of the IEO and UniL Technological Units for valuable support. We thank Chiara Soriani of the IEO Imaging Unit and Alessandro Cuomo of the IEO Proteomic Unit for technical support. We thank the ALEMBIC facility at San Raffaele Scientific Institute for the electron microscopy experiments. We thank Pierre Becquart of the Habib lab for technical help in reagent preparation. We are grateful to Edward M. De Robertis (UCLA) and Gary Davidson (Karlsruhe Institute of Technology) for sharing the LRP6 vectors. We thank Miguel Angel del Pozo and Fidel Lolo (CNIC Madrid) for sharing Cav1-expressing vectors. We thank Claudia Tredup of the Structural Genomics Consortium (Goethe University Frankfurt) for the generous gift of the CK1 inhibitor MU1742. We thank Evelyne Coudrier, Tsai Feng Chin, and Guerin Gwendal (Curie Institute, Paris) for the generous gift of the myosin1B antibody. We are grateful to Sergio Acebron (COSP Heidelberg), Nico Mitro (IEO), Sara Sigismund (IEO) and Stefano Santaguida (IEO) for valuable suggestions, and to all members of the Mapelli, Habib and Pasini laboratories for scientific discussion and carefully reading the manuscript. S.E., G.R., P.G., and F.C. were Ph.D. students within the European School of Medicine (SEMM). This work was supported by

grants to M.M. from the Italian Association for Cancer Research (AIRC) (IG 18629) and the Italian Ministry of Health, under the "PSC Salute, Traiettoria 4" program (CAL.HUB.RIA project, code T4-AN-09). S.T. was supported by Fondazione Cariplo (Cariplo Giovani 2020-3576). N.C.G. was supported by the Italian Association for Cancer Research (AIRC) (IG 27101). S.J.H. was supported by funds from the University of Lausanne. This was partially supported by the Italian Ministry of Health with Ricerca Corrente and 5 × 1000 funds.

## Author contributions

S.E. and G.R. performed the cell biology, imaging and biochemical experiments and analyzed the data. P.G., M.B., F.C., F.D., and C.G. performed molecular biology and biochemical experiments S.E. and F.R. established Wnt3a-bead pull-down protocols. A.L. performed electron microscopy acquisition. A.D. Mattia Marenda, S.R., Z.L., and D.P. assisted with the imaging studies. N.G. contributed to TIRF experiments. S.T. and F.I. performed proteomic experiments. J.L.A.S. and P.T. performed Wnt-mediated asymmetric cell division in mESCs. S.T., D.P., N.G., S.J.H., and M.M. supervised the project. S.J.H. and M.M. wrote the manuscript.

## Competing interests

The authors declare no competing interests.

## Additional information

[1]IEO, European Institute of Oncology IRCCS, Milan, Italy. [2]Department of Biomedical Sciences, University of Lausanne, Lausanne, Switzerland. [3]Department of Health Sciences, University of Milan, Milan, Italy. [4]IRCCS Ospedale San Raffaele, Experimental Imaging Center, Milan, Italy. [5]IFOM ETS, The AIRC Institute of Molecular Oncology, Milan, Italy. [6]Present address: AMOLF Institute, Amsterdam, The Netherlands. [7]These authors contributed equally: Susanna Eli, Greta Rauso. ✉e-mail: ShukryJames.habib@unil.ch; marina.mapelli@ieo.it

