## [Transparent Peer Review file · Nature Communications]

Localized Wnt-signaling promotes asymmetric NuMA-dependent oriented divisions and unequal apportioning of mitochondria

Corresponding Author: Dr Marina Mapelli

Version 0:

Reviewer comments:

Reviewer #1

(Remarks to the Author)

In this manuscript, Eli et al. set out to understand how localized Wnt signals alter spindle orientation in dividing cells in a NumA/Lrp6/B-catenin dependent manner. Using proteomics, they also identified enrichment of specific subsets of proteins, including that of mitochondria at the Wnt-proximal region of a dividing cell, and propose that localized Wnt signals lead to asymmetric partitioning of mitochondria. While a number of interesting experiments have been performed by the authors, there are some issues with analysis and several details in the experiments that have not been fully explored or explained. The mechanism underlying the asymmetric partitioning has also not been probed. For these reasons, I cannot recommend this article for publication in the current form.

My specific comments are:

1. Fig. 1a - this schematic can be combined with 1b since it does not add very much.

2. Fig. 1c - It would be ideal to mark the apparent metaphase plate in the images bounded by the red square (where the measurements of the angle beta were made).

There is a marked difference between the H2B-GFP signal in Wnt3a-bead images compared to BSA-bead images - the signal in the former appears much more diffuse and spread out compared to the latter. How were the positions of the metaphase plate determined in cases like this? This could be a significant source of error if not carried out in an automated fashion.

3. Fig. 1d, here and other such data obtained from cells in contact with beads, (eg. Fig. 1h, 2a, 2b, 2e) please add representative images.

4. In Fig. S1d/e; intensity along the membrane with respect to the bead position needs to be quantified. There appears to be much higher intensities of proteins at locations other than just at the bead. How would this be explained?

5. Fig. 2d: The MT extension to the Wnt3a bead is not convincing - the data using BSA beads as well as quantification of the number of MTs in contact with the bead needs to be performed.

6. It was unclear to me why Fig. 5f experiments were carried out on a Wnt3a platform, rather than using localized Wnt3a beads. Surely, this signal is no longer localized and not comparable to the data obtained in the other experiments? Live-cell 3D imaging should be feasible to visualize enrichment of myosin 1C in this case.

7. S4C: There doesn't appear to be an enrichment of MACF1 in the example images provided here.

8. P15, l 387: While the specific proteins involved in the spindle orientation in response to Wnt3a, and those responsible for alignment to the substrate have been independently identified in depletion experiments, the authors could also perform the depletions in the setting described in Fig. 1f (Wnt3a bead + signal) to understand how the force balance shifts in this system.

9. Fig. 6a,b: The mitochondrial density measurements at the proximal region need to be quantified better - the ratio of mitochondria in the proximal sector to total mitochondria in the cell needs to be compared against mitochondrial proportion in sectors of similar sizes across the entire cell.

10. Fig. 6d: It is hard to see mitochondria/ROI in the representative image .

11. Fig. 6f: The distribution of mitochondria relative to cell volume needs to be performed. Typically mitochondrial partitioning in mitosis scales with daughter cell volume, and might have nothing to do with the localized Wnt3a signal.

12. While the authors claim in the abstract that they demonstrate that NuMA/dynein/dynactin form a complex with Lrp6, they have neither visualized dynein/dynactin, nor depleted them to show specific involvement.

13. The title seems to suggest that the crux of the paper is 'unequal apportioning of mitochondria' mediated by localized Wnt signaling. However, very little of the paper actually described the mitochondrial partitioning. This aspect has not also been fully explored, and the mechanisms underlying the apparent mitochondrial asymmetry are unclear.

Reviewer #2

(Remarks to the Author)

The MS entitled "Localized Wnt-signaling promotes asymmetric NuMA-dependent oriented divisions and unequal apportioning of mitochondria" by Eli et al. examines the mechanisms by which Wnt signaling promotes asymmetric divisions. The authors use a technique with which Prof. Habib demonstrated that Wnt-tethered beads could directionally activate the Wnt signaling pathway in cultured cells and thereby regulate the direction of cell division. In this follow-up study, they employ a novel proteomic approach that relies on the magnetic property of the beads to identify protein enriched in the point of contact of the Wnt beads. In this study, they identify the NuMA protein as a key factor in this phenomenon and characterize the direct binding of this complex to the Wnt co-receptor LRP6. The MS thus adds a new layer of mechanism for Wnt regulation of asymmetrical cell division. Although the findings are interesting, some questions of mechanism should be addressed more fully, particularly regarding the understanding of the Wnt signaling mechanism in this phenomenon and the largely unexplained observation of asymmetrical mitochondrial accumulation:

1. It is not clear whether the transcriptional activation of Wnt signaling target genes plays a role in this context. Considering that multiple shRNAs used will lead to Wnt signaling activation or suppression, it would be important to resolve this question. shAPC and shAxin1, for example, will result in Wnt activation, and shLRP6 and sh β -catenin, will result in Wnt inhibition. Also, it is not clear whether β -catenin plays a transcriptional role (via TCF/LEF) or a direct role (by direct binding) in the spindle orientation.

2. shAPC and shAxin1 will activate Wnt signaling, while shLRP6 and shDvl2/3 will inhibit Wnt signaling transduction. Regarding Figure 2h, considering that shAPC didn't affect the spindle, does it suggest downstream Wnt signaling activation doesn't play a role? Can they confirm this? Considering that shAxin1 affects spindle orientation, does it imply an Axin direct role in the spindle axis orientation?

3. It is not clear whether LRP6 activation is required for the binding of the NuMA complex (Fig. 4b), nor are the relevant LRP6 motifs required for this binding clear. Considering the binding findings and the relevance of CK1 for the effect, is there any role of conserved intracellular domains of LRP6, such as the PPPSPxS motifs?

4. Is Wnt signaling activation required for the NuMA complex recruitment, such as in a cell without LRP6, and with LRP6 with mutated PPPSPxS motifs? In this context, LRP6 would be present, but there would be no Wnt signaling activation.

5. Depletion of Dvl1, Dvl2, or Dvl3 alone has been shown to affect the spindle axis. Why, in this context, does Dvl3 have no role? (PMID: 20823832). The rationale to focus on Dvl2 is unclear (is Dvl2 being used as a prototypical Dvl?). Please also explain the rationale not to evaluate the role of Dvl1 in this context.

6. Were Fzd receptor and LRP6 co-receptor enriched in the magnetic microbead proximity approach? What would be potential explanations for not identifying these receptors?

7. What is the rationale for using EGF beads as a control, rather than another ligand, such as Shh (considering that Frizzled1-10 receptors and Smoothed are both Class F GPCRs), or a non-functional Wnt3a?

8. In Fig. S1i, shAxin1 resulted in spindle randomization. However, in Fig. 1h, shAxin1 didn't affect spindle organization. What is the difference between the Substratum vs Bead experiment?

9. Others have observed asymmetric inheritance of mitochondria in stem cells, plant cells, and yeast – the observation that it can be triggered by a Wnt bead is, however, novel. Yet it is disappointing that there is no mechanistic understanding of how this comes about. The mitochondria do not appear to be in direct contact with the plasma membrane near the bead and so a direct interaction with the Wnt receptors is an unlikely explanation, especially for the proteins of the inner mitochondrial membrane they report. Is it a consequence of binding to astral microtubules and if so, why just those microtubules? Binding to local Actin? Directed trafficking or capture by a cytoskeletal element? This portion of the manuscript is strictly observational.

10. The bead pulldown experiment is not easy to interpret – what exactly is represented by the pulldown? Are some membranes (e.g. the mitochondrial membrane) still intact? Is the cytoskeleton largely intact? It seems unlikely that "protein network" or "complex of proteins" is an adequate description.

Minor issue:

There is an incomplete title in the Material and Methods section: " Lentiviral infection and "

Minor issue:

There are many grammar issues that will require careful editing.

A comment on statistics: Figure 1h displays three independent replicates and a total number of 362, 130, 22, 194, 187, 165, and 169 cells. Considering the extremely low p-values, such as <0.0001 , it is unclear whether statistics were performed using the three independent replicates averages or if the total number of cells was used, which could be interpreted as pseudoreplication. The same is true for other experiments throughout the MS.

Reviewer #3

(Remarks to the Author)

Version 1:

Reviewer comments:

Reviewer #2

(Remarks to the Author)

I appreciate the time the authors took to respond to the comments and the additional figures clarifying the non-role of transcription, the LRP6 mutations, and the CENPF/Miro knockdowns. I have no further concerns. Still many outstanding questions about how the mitoses are selectively trafficked to one pole over the other (is CENPF asymmetrically localized?), but that needn't be in this paper.

Reviewer #3

(Remarks to the Author)

Department of Experimental Oncology
European Institute of Oncology
Via Adamello 16, Milan
I-20139, Italy

Marina Mapelli, PhD
+39.02.94375018
marina.mapelli@ieo.it

Reviewer #2 (Remarks to the Author)

The MS entitled "Localized Wnt-signaling promotes asymmetric NuMA-dependent oriented divisions and unequal apportioning of mitochondria" by Eli et al. examines the mechanisms by which Wnt signaling promotes asymmetric divisions. The authors use a technique with which Prof. Habib demonstrated that Wnt-tethered beads could directionally activate the Wnt signaling pathway in cultured cells and thereby regulate the direction of cell division. In this follow-up study, employ a novel proteomic approach that relies on the magnetic property of the beads to identify protein enriched in the point of contact of the Wnt beads. In this study, they identify the NuMA protein as a key factor in this phenomenon and characterize the direct binding of this complex to the Wnt co-receptor LRP6. The MS thus adds a new layer of mechanism for Wnt regulation of asymmetrical cell division. Although the findings are interesting, some questions of mechanism should be addressed more fully, particularly regarding the understanding of the Wnt signaling mechanism in this phenomenon and the largely unexplained observation of asymmetrical mitochondrial accumulation:

We thank the Reviewer for the positive comments on the overall quality and significance of the work, we are delighted the manuscript has been well-received. We have addressed the points raised, including the molecular mechanism of mitochondrial polarization towards localized Wnt3 sources, as described below.

1. It is not clear whether the transcriptional activation of Wnt signaling target genes plays a role in this context. Considering that multiple shRNAs used will lead to Wnt signaling activation or suppression, it would be important to resolve this question. shAPC and shAxin1, for example, will result in Wnt activation, and shLRP6 and sh β -catenin, will result in Wnt inhibition. Also, it is not clear whether β -catenin plays a transcriptional role (via TCF/LEF) or a direct role (by direct binding) in the spindle orientation.

We thank the Referee for the comment, that gives us the opportunity to clarify. We agree that uncoupling the transcriptional Wnt response mediated by β -catenin from its transcription-independent mitotic activities is an important aspect for our studies.

As correctly noted by the Referee, in HeLa cell lines stably depleted of components of the Wnt signal transduction cascade, Wnt target gene transcription is altered in a way that might affect the mitotic events we are characterizing. Specifically, downregulation of LRP6 and β -catenin inhibits Wnt target gene transcription, while silencing of the destruction complex components APC and Axin1 results in β -catenin stabilization and hence constitutive Wnt-gene transcription activation. To uncouple nuclear β -catenin transcriptional activities from mitotic spindle orientation events, we designed a protocol to activate or inhibit Wnt target gene transcription in the interphase preceding the mitosis when we analyze orientation towards the Wnt3a-bead. To activate β -catenin-dependent Wnt transcription in shLRP6 HeLa cells, we synchronized cells in G1 by thymidine block, and after release we treated them with 10 μ M CHIR-99021. We then washed-out the drug and, when cells started entering into mitosis, we added Wnt3a-beads and siRNA for filming division orientation. Spindle orientation analyses confirmed that Wnt target gene transcription activation does not rescue the misorientation phenotype observed in shLRP6 HeLa cells. Importantly, activation of Wnt signaling in shCtrl HeLa cells with the same protocol, did not induce misorientation. To assess whether altered transcription in shAPC and shAxin1 HeLa cells in interphase could impact on the subsequent division orientation toward Wnt3a-beads, we first tested the expression levels of β -catenin and the mRNA levels of key Wnt target genes including TCF7 and RNF43. These analyses revealed that in HeLa cells stable APC depletion does not induce a significant β -catenin stabilization nor constitutive Wnt gene transcription activation, possibly because HeLa cells are aneuploid and can adapt to APC loss by activating negative feedback loops. This said, we decided to focus only on shAxin1 cells, which show a modest stabilization of β -catenin and Wnt target gene transcription activation. We inhibited β -catenin-dependent transcription in shAxin1 HeLa cells by

treatment with LF3, which disrupts β -catenin/TCF interaction (Fang, Cancer Research, 2016. doi: 10.1158/0008-5472.CAN-15-1519), and measured spindle orientation of shAxin1 treated cells towards Wnt3a-beads compared to DMSO treated cells. This experiment showed that Wnt-transcription inhibition does not rescue the misorientation phenotype induced by shAxin1 loss. These data are presented in a new *Supplementary Figure 2*. Based on these findings, we conclude that that transcriptional activation of Wnt signaling target genes does not play a role in the β -catenin spindle orientation activities, and that this process relies on previously uncharacterized mitotic β -catenin scaffolding functions.

These results are consistent with the notion that we are studying a mitotic process where we characterize the response to localized Wnt3 signals with a protocol in which Wnt3a-coated beads are seeded into HeLa cells about 3 hours before filming or performing IP experiments. During this period of localized Wnt3a activation, cells are mostly in mitosis and the nucleus already disassembled. Thus, it is unlikely that Wnt-bead signaling triggers gene significant transcription. What we show is that localized mitotic Wnt3a signaling acts in division orientation processes by inducing the assembly of cortical LRP6 with β -catenin and the spindle orientation protein NuMA at the cortical site in contact with the Wnt3-bead, where the Wnt-pathway is locally activated.

2. shAPC and shAxin1 will activate Wnt signaling, while shLRP6 and shDvl2/3 will inhibit Wnt signaling transduction. Regarding Figure 2h, considering that shAPC didn't affect the spindle, does it suggest downstream Wnt signaling activation doesn't play a role? Can they confirm this? Considering that shAxin1 affects spindle orientation, does it imply an Axin direct role in the spindle axis orientation?

We thank the Reviewer for this comment. The spindle orientation analyses towards Wnt3a-beads presented in Figure 1g describe a new Wnt-dependent mitotic mechanisms that involves cortically localized activation of the signaling pathway, and the assembly of a complex between some of the components participating in canonical Wnt-signaling and the spindle motors NuMA/dynein/dynactin. Consistently, silencing of β -catenin, LRP6, Dvl1 and Dvl2 significantly impairs orientation toward the Wnt3a-beads, suggesting that these proteins participate in the mechanism that aligns the mitotic spindle perpendicular to the bead. Wnt-bead pull-down experiments and the co-IPs of Figure 4 confirm that Dvl2, Lrp6 and β -catenin form a complex also with NuMA at the cortex in Wnt-ON conditions. In turn, data in Figure 1g show that APC is not essential for spindle orientation towards Wnt3a-beads nor Axin1 (they both have no statistically relevant spindle axis distribution compared to shCtrl HeLa cells), likely because they do not enter the β -catenin/NuMA complex.

As clarified in the reply to the previous point, this is consistent with the evidence that canonical Wnt signaling activation is not directly implicated in spindle orientation mechanistic.

3. It is not clear whether LRP6 activation is required for the binding of the NuMA complex (Fig. 4b), nor are the relevant LRP6 motif required for this binding clear. Considering the binding findings and the relevance of CK1 for the effect, is there any role of conserved intracellular domains of LRP6, such as the PPPSPxS motifs?

We are grateful to the reviewer for raising a valid point regarding the requirement of LRP6 activation for the formation of the NuMA/ β -catenin/LRP6 complex, we agree that this is a key point in the study. Experiments of Figure 4 show that the association of LRP6 with NuMA occurs only in Wnt-ON conditions, i.e. when cells are treated with Wnt conditioned medium and not when they are treated with DKK1. We reasoned that also the LRP6 co-receptor recruited at the Wnt3a-beads used in the bead-pulldown of Figure 4a is probably activated by being engaged with the Wnt3a beads. To confirm this hypothesis, we repeated the bead pull-down assay and showed that the LRP6 retained on the beads is, at least in part, phosphorylated on Ser1490, confirming that it is activated (new *Figure S5a*). To further prove the need of LRP6 activation for the interaction with NuMA, we generated an LRP6 mutant mutated in the five PPPSPxS motifs (LRP6-M5), which cannot be phosphorylated by CK1/GSK3 β and is unable to activate Wnt signaling (Niehrs, CMLS, 2010. doi: 10.1007/s00018-010-0329-3; Tamai, Mol Cell, 2004. doi:10.1038/nature04185; Davidson, Nature, 2005. doi:10.1038/nature04170; Zeng, Nature, 2005. doi:10.1038/nature04185). Co-IP experiments conducted in mitotic extracts of sh-LRP6 HeLa cells transfected with sh-resistant GFP-LRP6 constructs showed that LRP6-M5 does not interact with NuMA while LRP6 wild-type does. These data indicated that LRP6 activation by phosphorylation is required for the interaction with NuMA. These data are presented in the new *Figure 4c*.

4. Is Wnt signaling activation required for the NuMA complex recruitment, such as in a cell without LRP6, and with LRP6 with mutated PPPSPxS motifs? In this context, LRP6 would be present, but there would be no Wnt signaling activation.

We thank the Referee for the interesting comment. To integrate the biochemical experiments described in the answer to the previous point, we have performed cortical NuMA rescue experiments by exogenous expression of sh-resistant LRP6 wild-type or LRP6-M5 in shLRP6 HeLa cells. Measurements of cortical NuMA at the Wnt3a-bead contact site in shLRP6 mitotic HeLa cells expressing LRP6-WT or the LRP6-M5 mutant showed that only LRP6-WT can recruit NuMA at the site of Wnt activation, confirming that LRP6 activation is required for recruitment of microtubule motors and spindle orientation perpendicular to the Wnt3 source. These data are included in the new *Figure 4d-e*.

5. Depletion of Dvl1, Dvl2, or Dvl3 alone has been shown to affect the spindle axis. Why, in this context, does Dvl3 have no role? (PMID: 20823832). Kikuchi, EMBO J 2010. this is a different “orientation” to the plate, as shown in Fig S1k-l. The rationale to focus on Dvl2 is unclear (is Dvl2 being used as a prototypical Dvl?). Please also explain the rationale not to evaluate the role of Dvl1 in this context.

We thank the Reviewer for the insightful comment. In fact, spindle orientation mechanisms have been extensively studied in cells in isolation plated on fibronectin coated slides, which divide with the mitotic spindle axis parallel to the substrate. In this setting, it has been shown that Dvl2 and Dvl3 contribute to maintain the division parallel to the plate, as described in Kikuchi, EMBO J. 2010. This is confirmed also in our experiments (Supplementary Fig. S1n-o). Several studies showed that division orientation to the plate involves adhesion pathways and actin-based retraction fibers, stabilizing contacts of mitotic cells to the substrate (Lechler, Nat. Rev. Mol. Cell Biol. 2021;

doi:10.1038/s41580-021-00384-4). The molecular mechanism responsible for orientation orthogonal to the Wnt3-beads might only partly overlap with the one aligning the spindle to the substratum because, for instance, it does not need to crosstalk with adhesion pathways. Interestingly Dvl2 and Dvl3 have been shown to interact with mitotic NuMA by co-IP experiments in HeLa cells (Yang, PNAS 2014. doi: 10.1073/pnas.1319341111). This considered, we initially decided to test the implication of Dvl2 and Dvl3 in spindle orientation mechanisms towards the Wnt3a-beads. Intriguingly, we found that in Dvl2 is important for orientation to the Wnt3 source, while Dvl3 is not. To explain this result, we tested whether depletion of Dvl3 could be compensated for an increase in Dvl2, but this is not the case (shown in Supplementary Fig. S1). Thus, we think that the differential involvement of Dvl2 and Dvl3 in localized mitotic Wnt3 response can be ascribed to different mitotic functions of the two Dvl isoforms, with molecular details to be further investigated.

As suggested by the Referee, to assess if Dvl1, which was not previously implicated in spindle placement, is important for orientation toward Wnt3a-beads, we generated a HeLa cell line stably depleted of Dvl1 by lentiviral transduction, and monitored division orientation to the Wnt3a-bead by time-lapse videorecording. This experiment showed that Dvl1 contributes to position the mitotic spindle axis orthogonal to the Wnt3a-bead. These data have been added to *Figure 1g*.

6. Were Fzd receptor and LRP6 co-receptor enriched in the magnetic microbead proximity approach? What would be potential explanations for not identifying these receptors?

We thank the Referee for the question. We could not detect enrichment of Fzd receptors nor of the LRP6 co-receptor in the bead pulldown experiment by proteomic. We reason that this is not unusual in Mass-Spectrometry experiments considering that transmembrane proteins are difficult to solubilize and we are using mild lysis conditions in order to preserve protein complexes recruited at the bead contact sites as much as possible. However, immune-blot of bead pull-down experiments conducted with the same protocol used for the proteomic data, showed that LRP6 is recruited at the Wnt3a-bead contact site (Figure 4).

7. What is the rationale for using EGF beads as a control, rather than another ligand, such as Shh (considering that Frizzled1-10 receptors and Smoothed are both Class F GPCRs), or a non-functional Wnt3a?

We thank the Reviewer for asking about the rationale of the experiment conducted with EGF-coated beads. We agree that several controls could be considered to address the specificity of Wnt3 ligand compared to other signaling molecules. In choosing the control, we reasoned that a possible physiological setting for Wnt3a-dependent oriented division are intestinal stem cells residing in the bottom of intestinal crypts, which divide planarly with the spindle axis perpendicular to the Wnt3a-secreting Paneth cells. Crypt niche signals contributing to intestinal stem cell maintenance include Wnt3a but also EGF and Notch. For this reason, we decided to test whether localized EGF signaling activation could promote division orientation, which is not the case.

Additionally, as a specificity control for Wnt3a-bead orientation, we also used inactive Wnt3a-beads (iWnt3a) by treating Wnt3a-coated beads with the reducing agent DTT, which unfolds the

ligand by reducing the disulfide bridges. This experiment confirmed that functional Wnt3a ligands are required to promote division orientation perpendicular to the bead (shown in Figure 1c).

8. In Fig. S1i, shAxin1 resulted in spindle randomization. However, in Fig. 1h, shAxin1 didn't affect spindle organization. What is the difference between the Substratum vs Bead experiment?

We thank the Referee for the comment, that gives us the possibility to clarify the differences between the alignment to the plate and alignment towards the Wnt3a-bead. In general, mitotic spindle orientation in cultured cells is attained by the action of “force generators” assembled on NuMA/dynein/dynactin motors, which are anchored at restricted cortical sites by direct interactions with Gai and LGN. In this configuration, the retrograde movement of dynein on astral microtubules results in pulling forces towards the cortex which position the spindle.

Several studies showed that spindle alignment to the substratum depends mainly on adhesion mechanisms including integrins, focal adhesions, retraction fibers, and in general macromolecular complexes localizing at the plasma membrane in contact with the substrate, and is influenced by the organization of the spindle poles. In our current study, we show that Wnt3a-beads provide a stronger positional cue able to tilt the spindle axis toward the bead contact site. Mechanistically, we show that this is attained by recruitment of Gai/LGN/NuMA/dynein/dynactin at the Wnt3a-bead contact site by interaction with activated LRP6 coreceptors. Additional ancillary proteins participate in targeting or stabilizing microtubule motors at the Wnt3a-bead, as suggested by the proteomic studies and the Wnt3a-bead pull-down experiments. Based on the experiments shown in Figure 1g and Supplementary Figure S1n-o, it seems that both Dvl2 and Dvl3 play a role in orientation towards the substratum, but only Dvl2 is important for orientation towards the Wnt3a-bead. As we suggested in the response to point (5), additional investigations will be needed to uncover the molecular mechanisms underpinning this finding.

9. Others have observed asymmetric inheritance of mitochondria in stem cells, plant cells, and yeast – the observation that it can be triggered by a Wnt bead is, however, novel. Yet it is disappointing that there is no mechanistic understanding of how this comes about. The mitochondria do not appear to be in direct contact with the plasma membrane near the bead and so a direct interaction with the Wnt receptors is an unlikely explanation, especially for the proteins of the inner mitochondrial membrane they report. Is it a consequence of binding to astral microtubules and if so, why just those microtubules? Binding to local Actin? Directed trafficking or capture by a cytoskeletal element? This portion of the manuscript is strictly observational.

We are grateful to the Referee for appreciating that the description of mitochondrial polarization in response to Wnt3 sources and asymmetric apportioning between daughter cells is novel and interesting. We are aware that, in our initial submission, the mitotic crosstalk between mitochondria and Wnt signaling was described only phenomenologically. In order to elucidate the molecular mechanism underlying mitochondrial polarization towards Wnt3a-beads, we developed a protocol for assessing mitochondrial distribution in metaphase HeLa cells by quantitative imaging (new *Figure 6c-f*). This protocol was employed to study molecular factors necessary for inducing mitochondria polarization. Candidate proteins possibly implicated in Wnt-dependent polarization were selected from the Wnt3a-bead proteomic interactome, excluding proteins essential for mitotic

progression and division orientation (such as cortical F-actin binding proteins and microtubule binding proteins), or mitochondrial integrity. Based on these criteria, we decided to analyze the role of CENP-F, which has been implicated in mitochondrial movement to cell periphery (Kanfer, Nat Comm 2015, doi: 10.1038/ncomms9015), and Miro1-Miro2, two GTPases of the mitochondrial outer membrane acting as adaptors for mitochondrial transport on microtubules and actin cables (Oeding, JCS, 2018; doi: 10.1242/jcs.219469; López-Doménech, EMBO J., 2018; Kruppa, JCS, 2021; doi: 10.15252/embj.201696380). Measurement of mitochondria polarization towards Wnt3a-beads in HeLa cells depleted of CENP-F revealed that this protein is implicated in Wnt-dependent mitochondrial polarization toward the site of Wnt activation. Intriguingly, double depletion of Miro1 and Miro2 induces spindle misorientation and randomized distribution of mitochondria compare to the Wnt3a-bead position, likely because loss of both Miro proteins greatly reduces cell viability. These results are presented in *Figure 6g-i and Supplementary Figure S6g-j*.

10. The bead pulldown experiment is not easy to interpret – what exactly is represented by the pulldown? Are some membranes (e.g. the mitochondrial membrane) still intact? Is the cytoskeleton largely intact? It seems unlikely that “protein network” or “complex of proteins” is an adequate description.

We thank the Referee for raising the point regarding the bead pull-down proteomic experiment and the way we described it in the manuscript. The entire protocol to isolate protein complexes recruited at the cortex by Wnt3a-bead magnetic pull-down is novel, and required extensive optimization to retrieve components specifically associating with Wnt3a-beads versus BSA-beads. Basically, the protocol envisions beads seeding on metaphase-synchronized HeLa cells, lysis with mild detergents (0.4% NP40) in order to preserve complexes associating with the membrane in contact with the beads, and magnetic isolation without lysate clearing. The fact that mitotic extracts are not cleared by centrifugation implies that patches of membrane around the bead-immobilized Wnt3a ligands and Fzd/LRP6 engaged receptors are pulled-down. In fact, R-spondin, syndecan-1/4, caveolin1/2 and Wnt3a itself are found among the enriched hits (Figure 3b-c). Consistent with the imaging analysis and subsequent immunoblot experiments, also the membrane-associated G α i subunits and the cortical actin binding myosin-1B/C are among the top Wnt3a interactome hits, indicating that the cytoskeleton is not totally disrupted. Additionally, NuMA and CK1 α , which interact with G α i and the activated LRP6 co-receptor respectively, are enriched in the Wnt3a-bead interactome, suggesting that some non-covalent interactions among membrane-associated complexes are also maintained. These findings strongly support the conclusion that the bead pull-down proteomic approach can identify non-covalent complexes associating with the intracellular domains of transmembrane receptors engaged with ligands coating the magnetic beads, possibly including cytoskeleton binding proteins and mitochondrial proteins. Based on these considerations, we referred to the bead interactome as “protein network” in the first submission. To avoid misunderstanding, we have changed it to “protein complexes”.

Minor issue:

There is an incomplete title in the Material and Methods section: " Lentiviral infection and "

We thank the Reviewer for having noticed this typo. We have corrected it in the revised version to “Lentiviral transduction and transfection”.

Minor issue:

There are many grammar issues that will require careful editing.

We thank the Reviewer for the suggestion. We have edited carefully the manuscript to improve the text.

A comment on statistics: Figure 1h displays three independent replicates and a total number of 362, 130, 22, 194, 187, 165, and 169 cells. Considering the extremely low p-values, such as <0.0001 , it is unclear whether statistics were performed using the three independent replicates averages or if the total number of cells was used, which could be interpreted as pseudoreplication. The same is true for other experiments throughout the MS.

We thank the Referee for raising the point of the statistics we used to analyze spindle orientation data. As specified in the text, rose-plot distribution of spindle axis β -angles have been calculated from at least three biologically independent experiments, filming 30-100 cells for each condition. T-tests have been used to ensure that there was no statistical difference between replicates, before using the total number of cells to evaluate statistically significant differences. In our opinion, since each β -angle refers to the division orientation of an individual cell in isolation, which is not influenced by the rest of the population, we are inclined to think that no pseudoreplication occurs.

Reviewer #3 (Remarks to the Author)

We thank the Reviewer for having co-reviewed the manuscript and for the insightful comments and suggestions listed by Reviewer #1.

Reviewer #1 (Remarks to the Author)

In this manuscript, Eli et al. set out to understand how localized Wnt signals alter spindle orientation in dividing cells in a NumA/Lrp6/B-catenin dependent manner. Using proteomics, they also identified enrichment of specific subsets of proteins, including that of mitochondria at the Wnt-proximal region of a dividing cell, and propose that localized Wnt signals lead to asymmetric partitioning of mitochondria. While a number of interesting experiments have been performed by the authors, there are some issues with analysis and several details in the experiments that have not been fully explored or explained. The mechanism underlying the asymmetric partitioning has also not been probed. For these reasons, I cannot recommend this article for publication in the current form.

We thank the Reviewer for the comments. We recognize that additional clarifications on some experiments can be beneficial for a better understanding of the manuscript. We also are aware that, in our initial submission, the mitotic crosstalk between mitochondria and Wnt signaling was mainly observational. We have revised the manuscript according to the Referee's suggestions, explaining in more details the rationale and the outcome of the experiments, and providing the description of the molecular mechanism accounting for mitochondrial polarization in response to localized Wnt3 signals (*new Figure 6 and Supplementary Figure S6*), as outlined in the point-by-point reply below.

My specific comments are:

1. Fig. 1a - this schematic can be combined with 1b since it does not add very much. We thank the Referee for the suggestion. We have merged panel 1a and 1b in a new *panel 1a*.
2. Fig. 1c - It would be ideal to mark the apparent metaphase plate in the images bounded by the red square (where the measurements of the angle beta were made).

We appreciate the Referee's advice. To improve clearness, in the new *Figure 1b*, we have boxed the metaphase plate of the frame used for the β -angle measurements.

There is a marked difference between the H2B-GFP signal in Wnt3a-bead images compared to BSA-bead images - the signal in the former appears much more diffuse and spread out compared to the latter. How were the positions of the metaphase plate determined in cases like this? This could be a significant source of error if not carried out in an automated fashion.

We thank the Reviewer for this observation. The difference in the GFP signal at the DNA in the mounted frames depends on the focus of the filming acquisition, which might change in subsequent frames due to cell rounding. For each cell, the β -angle has been measured using the stack in which the metaphase plate is in focus. We have mounted a sequence of more focused frames for the BSA-bead experiment in the new *Figure 1b*.

3. Fig. 1d, here and other such data obtained from cells in contact with beads, (eg. Fig. 1h, 2a, 2b, 2e) please add representative images.

We thank the Referee for the suggestion. For the sake of completion, we have added a *Supplementary Figure S8* with representative metaphase frames of all the timelapse experiments of HeLa cells dividing in contact with coated beads, with the diverse genetic background and treatments analyzed in the manuscript.

4. In Fig. S1d/e = 2d/e; intensity along the membrane with respect to the bead position needs to be quantified. There appears to be much higher intensities of proteins at locations other than just at the bead. How would this be explained?

We thank the Reviewer for the comment, that gives us the opportunity to better explain the link between localized Wnt3a-bead signaling and cortical recruitment of microtubule motors

responsible for spindle positioning. The current model posits that mitotic spindle orientation in cultured cells is attained by the action of “force generators” assembled on NuMA/dynein/dynactin motors, which are anchored at cortical crescents at opposite site of the cell by direct interactions with Gai and LGN. In this configuration, the retrograde movement of dynein on astral microtubules results in pulling forces towards the center of the crescents whose resultant positions the spindle axis perpendicular to the crescent (Lechler & Mapelli, Nat. Rev. Mol. Cell Biol., 2021; doi: 10.1038/s41580-021-00384-4). What are the extracellular cues targeting dynein/dynactin motors at specific cortical sites in diverse cellular systems to instruct division orientation is not fully understood. In our current study, we show that Wnt3a-beads center one of the Gai/LGN/NuMA/dynein/dynactin crescent at the bead contact site, and that activated LRP6 coreceptors are key for this process (Figure 2b-c and S3c-e). According to the aforementioned spindle orientation mechanistic, centering of the microtubule motors in the middle of the cortical crescent suffice to generate astral microtubule pulling forces whose resultant instructs spindle axis positioning orthogonal to the bead. To quantify Wnt3a-beads position with respect to NuMA/LGN/Gai crescent, we plotted the intensity profiles of IF staining of these proteins in mitotic cells tracing a segmented line of 6 μ m from one end to the crescent to the opposite one, and overlaid them with the autofluorescence of the beads. These analyses confirmed that the Wnt3a-bead centers force generators at the site of Wnt signaling. These data have been added to Supplementary Figure S3f.

5. Fig. 2 d: The MT extension to the Wnt3a bead is not convincing - the data using BSA beads as well as quantification of the number of MTs in contact with the bead needs to be performed.

We thank the Referee for the suggestion. To confirm that astral microtubules reach out to the Wnt3a-bead to orient the mitotic spindle orthogonal to the bead, we quantified the intensity of astral microtubule in the proximity of Wnt3a-beads and compared it to that of astral microtubules in proximity of BSA-beads. These analyses confirmed that more astral microtubules reach the cortex at the Wnt3a-bead contact site, fully supporting the notion that dynein motors exert a traction force by retrograde movements towards the nearest spindle pole. These data are presented in a new Figure 2e-f.

6. It was unclear to me why Fig. 5f experiments were carried out on a Wnt3a platform, rather than using localized Wnt3a beads. Surely, this signal is no longer localized and not comparable to the data obtained in the other experiments? Live-cell 3D imaging should be feasible to visualize enrichment of myosin 1C in this case.

We thank the Reviewer for the comment concerning the experimental approach that we designed to monitor Myosin1C-dependent stabilization of Wnt3a-contact sites in mitotic cells. Specifically, the question we wanted to address is whether there is a local remodeling of the actomyosin cytoskeleton at the site of Wnt3a activation, as suggested by the enrichment of Myosin1C and Myosin1B in the Wnt3a-bead pull-down proteome. To this aim, we needed to follow in space and time the dynamics of the mitotic cortex right after localized Wnt activation. TIRF microscopy is the preferable option for these kind of studies as it restricts the excitation and detection of fluorophores to a thin region of the specimen supported by a glass slide, dramatically improving

the spatial resolution of the features or events of interest. We reasoned that performing TIRF experiments at the bead contact site would not be feasible for several reasons: in our protocol beads are seeded on cells before they enter in mitosis. Additionally mitotic HeLa cells round up making high-resolution dynamic imaging studies at the time of contact between the bead and the cortex very challenging. Previous studies have shown that Wnt3a-coated platform serves as a *basal localized cue* to promote asymmetric cell division of skeletal muscle cells perpendicular to the coated surface (Lowndes, Stem Cell Rep. 2016, doi: 10.1016/j.stemcr.2016.06.004; Okuchi, Nat. Mater., 2021; doi:10.1038/s41563-020-0786-5). Based on these considerations, to follow local actomyosin rearrangement at the site of Wnt activation we decided to film rounded metaphase HeLa cells landing on a Wnt3a-coated surface by TIRF microscopy, monitoring GFP-Myosin1C fluorescence as a proxy for contacts between cortical F-actin and the plasma membrane where the Wnt signaling gets activated. These experiments showed that localized Wnt signaling stabilizes the contacts between mitotic actomyosin and the plasma membrane, providing the rational for understanding the molecular basis of spindle positioning towards the Wnt3a-beads.

7. S4C: There doesn't appear to be an enrichment of MACF1 in the example images provided here.

We agree with the Referee's comment that there no specific enrichment of MACF1 is visible in the images of Figure S4c. We have removed panel.

8. P15, l 387: While the specific proteins involved in the spindle orientation in response to Wnt3a, and those responsible for alignment to the substrate have been independently identified in depletion experiments, the authors could also perform the depletions in the setting described in Fig. 1f (Wnt3a bead + signal) to understand how the force balance shifts in this system.

We thank the Referee for the comment and the suggestion of further exploring the crosstalk between orientation to the Wnt3a-beads and orientation to the substrate where cells are plated. To this aim, we repeated the α/g angle measurement experiments shown in Figure 1e-f (Figure 1f-g in the original submission) using HeLa cells stably depleted of DVL3, which induces spindle misorientation compared to the substrate but not to the Wnt3a-bead, or lacking β -catenin, which cannot orient towards the Wnt3a-bead but retain alignment to the plate. These experiments revealed that lack of DVL3 causes a significant misorientation with respect to the substratum (α angle) even in the presence of the Wnt3a-bead, whereas depletion of β -catenin does not alter significantly the α/g angles compared to sh-Ctrl HeLa cells, suggesting that it functions predominantly in orienting the mitotic spindle to the Wnt3a-bead. These data have been included in the new *Figure S1p*.

9. Fig. 6a,b: The mitochondrial density measurements at the proximal region need to be quantified better - the ratio of mitochondria in the proximal sector to total mitochondria in the cell needs to be compared against mitochondrial proportion in sectors of similar sizes across the entire cell.

We thank the Referee for the suggestion. We have quantified the mitochondrial density ratio between the proximal sector to the beads and the entire cell slice in the micrographs. These analyses confirmed that more mitochondria are found in proximity of Wnt3a-beads compared to BSA-beads. These data have been added to *Figure. S6a-b*.

0. Fig. 6d: It is hard to see mitochondria/ROI in the representative image.

We thank the reviewer for their comment. Following their advice, we have replaced the images and included a new one that highlights the ROI with and without the fences. We hope these updated visuals enhance the clarity and overall understanding.

1. Fig. 6f: The distribution of mitochondria relative to cell volume needs to be performed. Typically mitochondrial partitioning in mitosis scales with daughter cell volume, and might have nothing to do with the localized Wnt3a signal.

We thank the reviewer for this comment. Our analysis and quantification of TOM20 mean intensity take into account the volume of the cells. To clarify this, we have revised the *Materials and Methods* section and included a new supplementary figure (*Supplementary Figure S7*) illustrating the approach.

For the reviewer's convenience, instead of presenting the percentage distribution of mitochondria between the two daughter cells, we compiled data from doublet cells across all biological replicates. Using the same volume-corrected quantification of TOM20 mean intensity, we plotted all doublets in contact with either active Wnt beads or inactive Wnt beads (see below *Figure-Revision A and C*). The results show that, on average, the TOM20 mean intensity is higher in the Wnt-proximal cell compared to the Wnt-distal cell (*Figure-Revision B*). This difference is not observed in doublets in contact with inactive Wnt beads (*Figure-Revision D*).

Figure-revision: Mitochondria apportioning in mESCs. (A) Paired TOM20 mean intensity measurements for doublet cells in contact with Wnt3a-beads. (B) Average TOM20 mean intensity for doublet cells in contact with Wnt3a-beads. (C) Paired TOM20 mean intensity measurements for doublet cells in contact with (inactive) iWnt beads. (D) Average TOM20 mean intensity for doublet cells in contact with iWnt beads. Asterisks indicate statistical significance calculated by t-test for A-D: ns, not significant; * $P < 0.05$; ** $P < 0.01$

2. While the authors claim in the abstract that they demonstrate that NuMA/dynein/dynactin form a complex with Lrp6, they have neither visualized dynein/dynactin, nor depleted them to show specific involvement.

We thank the Reviewer for the comment, that gives us the possibility of explaining the rationale for the statement that NuMA/dynein/dynactin form complex with LRP6. Based on the current knowledge of spindle orientation mechanisms, NuMA acts as a dynein activating adaptor for spindle orientation mechanisms by physically sandwiching the N-terminal portion of its coiled-coil between dynein and dynactin (Pirovano, Nat Comm, 2019, doi: 10.1038/s41467-019-09999-w; Aslan, bioRxiv, 2024, doi: 10.1101/2024.11.26.625568; Colombo, JCB, 2025, doi: 10.1083/jcb.202408118). Consistent with this line of thoughts, the dynactin subunit p150-Glued was found enriched with NuMA in the Wnt3a-bead pull-down (Figure 4a). On these premises, monitoring the presence of NuMA at the cortex has often been considered sufficient to demonstrate that the NuMA/dynein/dynactin complex was there. To further support the notion that dynein/dynactin are at the cortex with NuMA in our experiments, following the Reviewer's suggestion, we quantified the presence of p150-Glued at the bead contact site in mitotic HeLa cells, and confirmed that dynactin is enriched at the Wnt3a-beads compared to BSA-beads. These data have been included in *Figure 2b-c*.

3. The title seems to suggest that the crux of the paper is 'unequal apportioning of mitochondria' mediated by localized Wnt signaling. However, very little of the paper actually described the mitochondrial partitioning. This aspect has not also been fully explored, and the mechanisms underlying the apparent mitochondrial asymmetry are unclear.

We thank the Reviewer for the observation. We are aware that, in our initial submission, the mitotic crosstalk between mitochondria and Wnt signaling was described only phenomenologically. In order to elucidate the molecular mechanism underlying mitochondria polarization towards Wnt3a-beads, we developed a protocol for assessing mitochondrial distribution in metaphase HeLa cells by quantitative imaging (new *Figure 6c-f*). This protocol was employed to investigate the specific molecular factors necessary for inducing this polarization. Candidate proteins possibly implicated in Wnt-dependent polarization were selected from the Wnt3a-bead proteomic interactome, excluding proteins essential for mitotic progression and division orientation (including cortical F-actin binding proteins and microtubule binding proteins), or mitotic mitochondrial integrity. Based on these criteria, we decided to analyze the role of CENP-F, which has been implicated in mitochondrial movement to cell periphery (Kanfer, Nat Comm 2015, doi: 10.1038/ncomms9015), and Miro1-Miro2, which are two GTPases of the mitochondrial outer membrane acting as adaptors for mitochondrial transport on microtubules and actin cables (Oeding, JCS, 2018; doi: 10.1242/jcs.219469; López-Doménech, EMBO J., 2018; Kruppa, JCS, 2021; doi: 10.15252/emj.201696380). Measurement of mitochondria polarization towards Wnt3a-beads in HeLa cells depleted of CENP-F revealed that these proteins are implicated in Wnt-dependent mitochondrial polarization. Depletion of CENP-F impair mitochondria polarization toward the Wnt3a-bead (*Figure 6g-i*). Intriguingly, we found that double depletion of Miro1 and Miro2 randomizes the division axis toward the Wnt3a-bead and leads to defective cell viability. Consistent

with these observations, the mitochondrial proximal-to-distal density ratio is also randomized (shown in *Figure S6g-j*). Importantly, these results provide the molecular mechanism whereby mitochondria polarize towards the Wnt3a-beads in mitosis as requested by the Referee. These data are presented in *Figure 6g-i and Supplementary Figure S6g-j*.

Department of Experimental Oncology
European Institute of Oncology
Via Adamello 16, Milan
I-20139, Italy

Marina Mapelli, PhD
+39.02.94375018
marina.mapelli@ico.it

October 8th, 2025

Dear Reviewers,

please find enclosed the response to the additional comments of Reviewer 2 to the manuscript ‘Localized Wnt-signaling promotes asymmetric NuMA-dependent oriented divisions and unequal apportioning of mitochondria’ submitted to Nat. Comm. (NCOMMS-24-68160A).

We have revised the text and the figures of the manuscript according to the Referee’s suggestions, as detailed below in the point-by-point answers to the Referee’s comments.

I thank you in advance for assessing the manuscript, and look forward to hearing from you.

Reviewer #2 (Remarks to the Author):

I appreciate the time the authors took to respond to the comments and the additional figures clarifying the non-role of transcription, the LRP6 mutations, and the CENPF/Miro knockdowns. I have no further concerns. Still many outstanding questions about how the mitos are selectively trafficked to one pole over the other (is CENPF asymmetrically localized?), but that needn't be in this paper.

We thank the Reviewer for acknowledging the improvements of the revised manuscript, particularly regarding the molecular mechanism underlying the division orientation and mitochondrial polarization in response to localized Wnt3a activation. We are aware that our findings open several interesting questions, which we intend to pursue in future follow-up studies.

Reviewer #3 (Remarks to the Author):

We thank the Reviewer for agreeing on the positive evaluation of the manuscript.

ADDITIONAL COMMENTS

We asked Reviewer #2 to comment on your response to Reviewer #1. Reviewer #2 stated that there were a few remaining concerns:

1. Regarding reviewer #1 comment #7, reviewer #2 stated that there was an apparent discrepancy between the figure and text. Reviewer #2 noted that while the text cites Figures 3a,b as evidence for MACF1 enrichments at the bead site but that there is no enrichment of any significance according to 3a and that the effect in 3b is small.

We thank the Reviewer for raising this point. At pg. 12 of the current version of the manuscript, we stated that “MACF1, enriched at the Wnt3a-bead contact site (Fig. 3b-c), was previously...”. With this sentence we wanted to refer to the volcano plot of the bead proteomic experiment shown in Fig. 3, in which MACF1 is indeed only marginally enriched at the Wnt3a-beads (Fig. 3b), and at the graph of the MS/MS counts for the individual replicates of the mass spectrometry experiment (Fig. 3c) showing that MACF1 is preferentially found in the Wnt3a-bead proteome. We agree with the Reviewer that the enrichment is small. Nonetheless, we could validate it by beads pull-down experiments followed by immune-blot as shown in Fig. 5a.

2. Reviewer #2 noted a concern regarding Supplementary Figure 6a,b. Reviewer #1 requested a comparison between bead-oriented wedges vs comparable wedges across the rest of the cell as opposed to bead-oriented wedges vs everything else. Reviewer #2 noted that this may mask uneven distribution of the mitochondria and doesn't explain why control beads cause an enrichment (albeit less than Wnt beads). Reviewer #2 also noted that this figures and the main figure 6 showed colored organelles and data points but that this was not adequately explained in the legend.

We thank Reviewer 2 for the interesting comment. To rule out possible effects of uneven distribution of the mitochondria in the metaphase cells analyzed in Fig. 6a-b and S6a-b, we have quantified the ratio of the mitochondrial density to total mitochondrial density across wedges of comparable area covering the entire cell area, for cells dividing in contact with Wnt3a-beads and BSA-beads. This analysis revealed that in cells dividing in contact with Wnt3a-beads, the mitochondrial density peaks in the wedge proximal to the Wnt3a-bead contact site, while the mitochondrial density is similar between wedges in cells in contact with BSA-beads, with a marginal increase in the wedge more distal to the bead (labeled +5 in Fig. S6b). The results are shown in the current Figure S6a-b. To better explain how the quantifications of the mitochondrial

density were performed and illustrated, we revised the legends of *Figure 6a-b* and *Figure S6a-b* describing the indicated panels.

3. Reviewer #2 noted an additional minor concern that in figure 6a (bottom row, BSA beads) the enlarged view in the right hand panel is not properly aligned with the red box in the left hand panel.

We thank the Reviewer for noticing the mistake. We have repositioned the red box in the bottom-left panel of Fig. 6a (showing the cell in contact with the BSA bead) to better align it with the close-up view on the right-hand side. This correction is included in the revised *Figure 6a*.